# Heterogeneity of neuroendocrine transcriptional states in metastatic small cell lung cancers and patient-derived models

Delphine Lissa[1,10], Nobuyuki Takahashi[2,3,10], Parth Desai[2], Irena Manukyan [4], Christopher W. Schultz[2], Vinodh Rajapakse[2], Moises J. Velez[5], Deborah Mulford[6], Nitin Roper[2], Samantha Nichols[2], Rasa Vilimas[2], Linda Sciuto[2], Yuanbin Chen[7], Udayan Guha [8], Arun Rajan [8], Devon Atkinson [9], Rajaa El Meskini [9], Zoe Weaver Ohler [9] & Anish Thomas [2✉]

Molecular subtypes of small cell lung cancer (SCLC) defined by the expression of key transcription regulators have recently been proposed in cell lines and limited number of primary tumors. The clinical and biological implications of neuroendocrine (NE) subtypes in metastatic SCLC, and the extent to which they vary within and between patient tumors and in patient-derived models is not known. We integrate histology, transcriptome, exome, and treatment outcomes of SCLC from a range of metastatic sites, revealing complex intra- and intertumoral heterogeneity of NE differentiation. Transcriptomic analysis confirms previously described subtypes based on *ASCL1*, *NEUROD1*, *POU2F3*, *YAP1*, and *ATOH1* expression, and reveal a clinical subtype with hybrid NE and non-NE phenotypes, marked by chemotherapy-resistance and exceedingly poor outcomes. NE tumors are more likely to have *RB1*, *NOTCH*, and chromatin modifier gene mutations, upregulation of DNA damage response genes, and are more likely to respond to replication stress targeted therapies. In contrast, patients preferentially benefited from immunotherapy if their tumors were non-NE. Transcriptional phenotypes strongly skew towards the NE state in patient-derived model systems, an observation that was confirmed in paired patient-matched tumors and xenografts. We provide a framework that unifies transcriptomic and genomic dimensions of metastatic SCLC. The marked differences in transcriptional diversity between patient tumors and model systems are likely to have implications in development of novel therapeutic agents.

[1] Laboratory of Human Carcinogenesis, Center for Cancer Research, NCI, Bethesda, MD 20892, USA. [2] Developmental Therapeutics Branch, Center for Cancer Research, NCI, Bethesda, MD 20892, USA. [3] Medical Oncology Department, Center Hospital, National Center for Global Health and Medicine, Tokyo, Japan. [4] Laboratory of Pathology, Center for Cancer Research, NCI, Bethesda, MD 20892, USA. [5] Department of Pathology, University of Rochester Medical Center, Rochester, NY 14642, USA. [6] Department of Medicine, University of Rochester Medical Center, Rochester, NY 14642, USA. [7] Cancer and Hematology Centers of Western Michigan, Grand Rapids, MI, USA. [8] Thoracic and GI Malignancies Branch, Center for Cancer Research, NCI, Bethesda, MD 20892, USA. [9] Center for Advanced Preclinical Research, Leidos Biomedical Research, Inc, Frederick National Laboratory for Cancer Research, Frederick, MD 21702, USA. [10] These authors contributed equally: Delphine Lissa, Nobuyuki Takahashi. ✉email: anish.thomas@nih.gov

Cancers that appear morphologically similar often have markedly different clinical features, respond variably to therapy, and have a range of outcomes. Tumor genomic profiling has led to the identification of previously unrecognized cancer subtypes, reflecting the biology and developmental origins of cancer. Treatments based on molecular subtypes have substantially transformed the care of patients with some cancers, notably including non-small cell lung cancer (NSCLC) marked by significant declines in mortality[1]. However, in other cancers such as small cell lung cancer (SCLC), the identification of molecular subtypes has remained an elusive goal. SCLC is an exceptionally lethal malignancy that accounts for 13% of all lung cancer with >30,000 new cases/year in the United States alone[1,2]. In the absence of clinically relevant molecular subgroups, SCLC lacks effective targeted therapies and is treated as a homogeneous disease with a one-size-fits-all approach.

The limited understanding of molecular subtypes in clinical SCLC samples stands in contrast to the considerable intertumoral morphologic and immunohistochemical heterogeneity that has been recognized in SCLC models for decades[3]. A coherent molecular explanation for this heterogeneity was recently proposed, classifying SCLC into high and low neuroendocrine subtypes based on relative expression of lineage-determining transcription factors[4]. While the consensus nomenclature provides an important starting point for classifying SCLC, several critical questions remain. A major unanswered question is whether the proposed subtypes – defined using human and murine SCLC cell lines and limited number of primary tumor samples – can robustly classify metastatic tumors, representing the majority of SCLC cases. This is important because most SCLCs have metastasized outside the chest at diagnosis[5], and only about 5% of cases are diagnosed at earlier stages and undergo resection[6]. Other open questions include the relationship between molecular subtypes and clinical features, the extent of inter- and intratumoral heterogeneity, plasticity between subtypes, and whether subtypes engender specific therapeutic vulnerabilities. Moreover, we have a limited understanding of the degree to which patient-derived models accurately recapitulate the distribution of subtypes seen in patients. A major barrier to clinical validation of the proposed subtypes is the limited availability of high-quality tumors for comprehensive molecular analyses. SCLC is often diagnosed using fine needle aspirates, and biopsies at relapse are not standard. Research biopsies are difficult to obtain due to rapid cancer progression and patient comorbidities. Underscoring this challenge, despite being a recalcitrant cancer with exceedingly poor outcomes, SCLC is not represented in large-scale sequencing efforts such as The Cancer Genome Atlas (TCGA).

Here, we evaluated SCLC biopsies from a range of metastatic sites and sought to determine the impact of molecular characteristics on SCLC phenotypes, providing a foundational resource of 100 small cell tumors, integrating histology, transcriptome, exome, treatment responses, and outcomes. Our analyses provide a coherent portrait of the molecular subclasses of metastatic SCLC, revealing intra- and intertumoral heterogeneity, and identify a subtype characterized by chemotherapy-resistance, with clinical implications. We also determine potential therapeutic vulnerabilities exposed by NE differentiation that could be advanced for clinical evaluation to optimize patient outcomes, and to rationalize prospective subtype-specific clinical trials.

## Results

**Patients**. We evaluated 100 small cell neuroendocrine cancers (SCNC) acquired by biopsies from 72 patients, including 62 patients with SCLC and 10 patients with extrapulmonary small cell cancer (EPSCC), by whole exome sequencing (WES) with matched normal DNA, RNA sequencing (RNA-seq) and immunohistochemistry (IHC) (Supplementary Table 1, Fig. 1a and Supplementary Fig. 1). EPSCCs – aggressive neuroendocrine tumors that arise de novo or due to lineage plasticity under selective pressure of targeted therapies[7] – were included given convergent transcriptional and epigenetic programs[8,9], and similarities to SCLC histology and clinical course[10]. The tumor samples – 88 SCLCs and 12 EPSCCs – underwent central histopathologic review confirming small cell carcinoma and expression of neuroendocrine markers (Supplementary Table 2).

The median patient age at diagnosis was 62 years (range: 29–86). Fifty-four (75%) patients were diagnosed with extensive-stage disease. All patients received first line platinum-based chemotherapy, and in most cases (44/72, 61.1%) the tumor was platinum-resistant, i.e., recurred within 90 days of first line chemotherapy. Most patients had enrolled on clinical trials of immunotherapy and DNA damage response-targeted agents (64/72, 88.9%)[11–14] at relapse. The median time from diagnosis to tumor sampling was 8 months (range: 0–47) and a median of two systemic therapies (range: 1–6) were administered previously. Most tumors were obtained at relapse (82/100), 15 tumors at diagnosis, and two during autopsy, and included 91 metastases and nine primary tumors. A single tumor was available for 48 (66.7%) patients and sequential tumors for 11 (15.3%) patients, with a median of 41 days (range: 3–645) between biopsies. Eight (11.1%) patients had multiple tumors sampled at the same time-point. Tumor sites represented included liver (29%), lymph nodes (24%), and lung (16%). Detailed patient clinical characteristics are available in Supplementary Table 3.

**Heterogeneity of neuroendocrine differentiation between SCNCs**. While SCLC is defined by neuroendocrine differentiation, a subset of SCLCs are characterized by reduced or lack of expression of neuroendocrine markers[3]. Using previously published gene signatures of neuroendocrine activity (Supplementary Table 4)[15–17], SCNCs were distributed across a continuum of neuroendocrine gene expression, with two main categories; a larger group defined by high expression of neuroendocrine genes (NE) and a smaller group with low expression of NE genes (non-NE) (Fig. 1b). Notably, despite little overlap among gene sets (Supplementary Fig. 2), expression of the different signatures strongly correlated with each other (Fig. 1c). We used the 50-gene signature and applied single sample gene set enrichment analysis (ssGSEA) to annotate each tumor as NE or non-NE. The score ranges from −1 to 1, with positive and negative scores respectively indicating NE and non-NE differentiation – a lower negative score providing more confidence that neuroendocrine differentiation is lacking[15]. Using this approach, most of the tumors were classified as NE (65/100, 65%), including 67% (59/88) of the SCLC and 50% (6/12) of EPSCC. At a global level, the projection of RNA-seq data onto an unsupervised principal component analysis (PCA)[8] revealed distinct clustering between NE and non-NE tumors (Fig. 1d). NE tumors strongly converged toward neuroendocrine prostate cancer, whereas non-NE tumors bordered lung adenocarcinoma on the trans-differentiation trajectory from adenocarcinoma to small cell cancer. SCLC and EPSCC clustered together underscoring the similarities between the tumor types (Supplementary Fig. 3).

NE and non-NE tumors exhibited morphological features reminiscent of SCLC cell lines with classical and variant features, respectively[18]. NE tumors consisted mostly of small cells with high nuclear-cytoplasmic ratios, finely granular chromatin distributed throughout the nucleus, and inconspicuous nucleoli. Non-NE tumors had relatively larger cells with moderate amounts of eosinophilic cytoplasm, one or more prominent

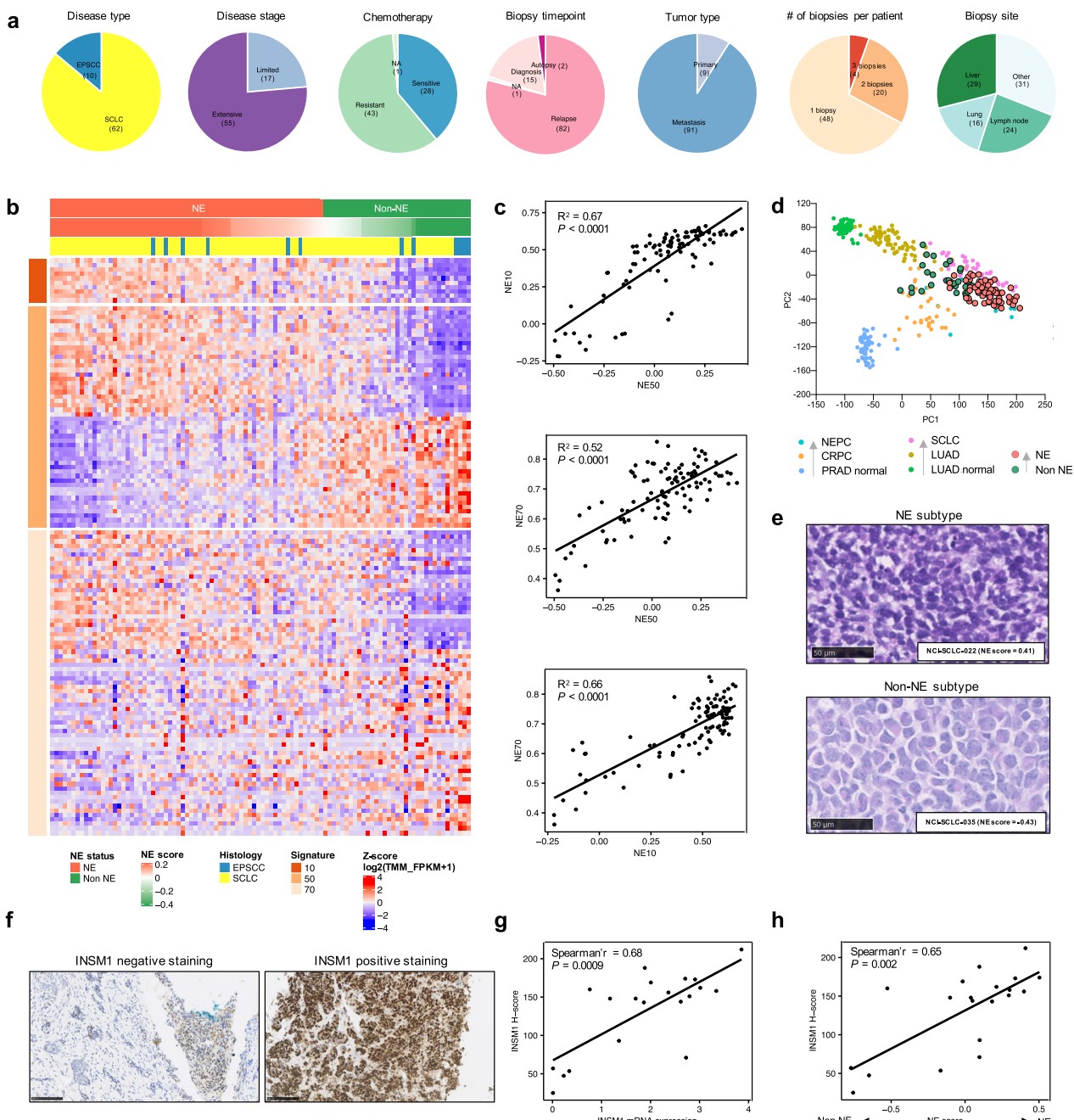

**Fig. 1 Neuroendocrine differentiation defines distinct SCNC subtypes. a** Pie charts summarizing patient and biopsy characteristics. **b** Heatmap of the 10-gene[16] (top panel), 50-gene[15] (middle panel), and 70-gene (lower panel)[17] neuroendocrine signatures. The 50-gene signature was derived from differentially expressed genes between matched normal adrenal cortex and medulla, 25 genes each correlating positively and negatively with neuroendocrine differentiation. The 70-gene and 10-gene signatures were derived from resistant prostate cancers with small cell or neuroendocrine features. Neuroendocrine scores and subtypes (NE or non-NE) derived from the 50-gene signature, and histology are indicated above the heatmap. **c** Pearson correlation between the three neuroendocrine signatures. R-squared values and the P-values are indicated (P < 2.2e-16). **d** Projection of 100 SCNC tumors onto the PCA developed by Balanis et al.[8], to evaluate the degree of neuroendocrine differentiation (trajectory indicated by arrows). **e** Representative photomicrograph images of H&E-stained small cell lung cancer of NE and non-NE subtypes. Black bars represent 50 µm (observations were repeated independently two times). **f** Representative images of IHC staining for INSM1 (observations were repeated independently two times). **g** Spearman correlation between INSM1 mRNA level and INSM1 H-score (n = 20 tumors). **h** Spearman correlation between 50-gene neuroendocrine signature score and INSM1 H-score (n = 20 tumors). All tests are two-tailed. Abbreviations: NE Neuroendocrine differentiation; SCLC Small cell lung cancer; EPSCC Extrapulmonary small cell cancer; TMM Trimmed Mean of M-values; FPKM Fragments Per Kilobase of Exon Per Million Fragments Mapped; H&E Hematoxylin and Eosin; NEPC neuroendocrine prostate cancer; CRPC castration-resistant prostate cancer; PRAD prostate adenocarcinoma; LUAD lung adenocarcinoma; PCA principal component analysis; NA not assessed.

nucleoli, and an open chromatin configuration with peri-nucleolar clearing (Fig. 1e). Nuclear expression of INSM1, a super-enhancer-associated transcription factor that regulates global neuroendocrine gene expression[19], was positively correlated with *INSM1* gene expression and the neuroendocrine score (Fig. 1f–h). Synaptophysin and chromogranin, membrane glycoprotein markers of neuroendocrine differentiation[20], were also more frequently expressed in NE tumors (Supplementary Fig. 4). Together these findings reveal the robustness of disparate neuroendocrine signatures to classify SCNCs to distinct NE and non-NE phenotypes, with substantial heterogeneity of neuroendocrine features between tumors.

**Intratumoral heterogeneity of SCNC neuroendocrine differentiation**. Although SCLC models show evidence of transcriptionally heterogeneous NE and non-NE cell populations[21–24], the extent of intratumoral heterogeneity in metastatic SCLC patient tumors is poorly understood, hindered by the lack of biopsy specimens. We sought to quantify the abundance of neuroendocrine cells in individual tumors. Overall, the relative proportion of neuroendocrine cells predicted by CIBERSORT [25] was concordant with the grouping determined by the 50-gene signature[15] (Fig. 2a). Yet, there was substantial variation in the predicted proportion of NE and non-NE cells within each tumor ranging from 45–100% and 48–100%, respectively, in NE and non-NE tumors. Heterogeneity was evident morphologically in some cases with variant-like tumor cells in a background of cells with classical features (Fig. 2b). Similar results were obtained when SCLC subtype-specific gene signatures[26,27] were applied, with varying subtype proportions noted within each tumor (Supplementary Fig. 5).

The scarcity of SCLC tissue specimens has led to cell line and mouse models from biopsies and circulating tumor cells (CTC) being used to interrogate SCLC biology[21,22,28,29]. While biopsy and CTC-derived xenograft (PDX and CDX) models are reported to capture the mutational landscape and functional features of the patient tumors, whether the models recapitulate the intratumoral heterogeneity of patient tumors is not known. We sought to put in context the magnitude of heterogeneity in our cohort and to compare intratumor heterogeneity across patient tumors and model systems. Metastatic and relapsed tumors from our cohort had relatively lower proportion of NE cells compared with early-stage, treatment-naïve SCLCs[30] (median proportion of NE cells: 58.0% and 64.8%, respectively; Fig. 2c, Supplementary Fig. S6a), indicative of decreased neuroendocrine differentiation with tumor progression[31] and chemotherapy[32]. In contrast, PDXs, CDXs[22,28], and cell lines[33,34] harbored markedly higher proportion of NE cells than patient tumors (median proportion of NE component in CDX, PDX/CDX, cell lines: 94.5% vs. 93.0% vs. 89.0%; Fig. 2c, Supplementary Fig. 6a). Accordingly, neuroendocrine scores of patient tumors were significantly lower than those of model systems (Supplementary Fig. 6b), which were enriched with NE tumors (Supplementary Fig. 6c). To further investigate the differences in heterogeneity of neuroendocrine differentiation between PDX models and patient tumors, we generated PDX models from patients with SCLC who underwent tumor biopsies at relapse, and performed RNA-seq on patient-matched tumor biopsy and xenograft tumor at the first passage (Supplementary Table 5). Notably, PDX tumors showed significantly higher proportion of NE cells (Fig. 2d, e) and neuroendocrine scores (Supplementary Fig. 6d), compared with the corresponding patient tumors. Of note, tumor purity was positively correlated with the neuroendocrine score, suggesting that the paucity of tumor microenvironment (TME) may partly account for the

higher neuroendocrine score observed in PDX tumors compared with patient tumors (Supplementary Fig. 7a, b).

Sections of the same tumor and longitudinal tumors obtained at multiple timepoints during the treatment course had similar neuroendocrine scores and estimated proportion of NE/non-NE cells (Supplementary Fig. 7c–e), suggesting overall stability of the intratumoral phenotype. Variations were mostly noticeable in tumors from different biopsy sites, indicative of inter-tumor heterogeneity. It is to be noted that sequential biopsy samples in this cohort were for the most part obtained at close intervals, and only 4/11 cases were obtained more than 2 months apart. Together these findings reveal substantial heterogeneity of neuroendocrine features within individual tumors and marked differences in transcriptional diversity between patient tumors and model systems.

**Neuroendocrine heterogeneity and expression of transcriptional regulators**. The recently proposed consensus nomenclature classifies SCLC based on expression of lineage-defining transcription factors ASCL1, NEUROD1, POU2F3, and YAP1 (SCLC-A, -N, -P, and -Y respectively)[4]. We found higher expression of *ASCL1* and *NEUROD1* in tumors classified as NE, consistent with the important role of these transcription factors in regulating neuronal and neuroendocrine differentiation[35] (Fig. 2f). Expression of *YAP1*, a transcription factor regulated by the Hippo signaling pathway was higher in tumors with non-NE differentiation[36], similar to other components of the Hippo pathway (Supplementary Fig. 8a). *POU2F3*, a master regulator of tuft cell identity[37] was upregulated in two non-NE tumors from a single patient and expressed multiple tuft cell lineage markers including *AVIL* and *IGF1R* (Supplementary Fig. 8a).

An unsupervised hierarchal clustering based on expression of the four subtype-defining transcription factors (Fig. 2g, Supplementary Fig. 8b) identified three major clusters corresponding to SCNC-A (cluster 1), SCNC-N (cluster 2) and SCNC-Y/-P (cluster 3). The tumor categorization by transcription factor expression closely aligned with the neuroendocrine signature-based classification (88.9% agreement with Cohen's kappa of 0.73). In addition, 19/24 (79%) biopsies obtained from the same patient were assigned to the same cluster (Supplementary Fig. 8b, c). The few discrepant cases either had comparable expression of all four genes, or co-expression of *ASCL1* and *YAP1*, as previously described[38]. Given that the 50-gene signature may be more robust to noise, when clustering was discordant with the NE subtype, tumors were re-classified using expression of the transcription factor characterizing the subtype with the greatest relative overall expression. Thus, most tumors were classified as SCNC-A ($n = 30$, 41.7%), followed by SCNC-Y ($n = 21$, 29.2%), SCNC-N ($n = 20$, 27.8%) and SCNC-P ($n = 1$, 1.4%). *ASCL1* was frequently co-expressed with *NEUROD1*, while the expression of *POU2F3* was mutually exclusive of the other transcription factors, as previously described[4,22,38]. A supervised PCA of the expression of the four genes revealed three clusters associated with the distinct molecular subtypes, further supporting subtyping of SCNCs into these molecular categories (Fig. 2h, Supplementary Fig. 8d). Three tumors classified as SCNC-N and Y demonstrated high expression of *ATOH1* and its downstream target *POU4F3* (Supplementary Fig. 9a). Genes differentially expressed in *ATOH1*-high tumors were enriched for pathways related to hair-cell and mechanoreceptor differentiation[22,39] (Supplementary Fig. 9b, c).

To further characterize the distinct transcriptional features of the subtypes, we evaluated the top 2000 differentially expressed genes across SCNC-A, -N, and -Y tumors. A supervised PCA revealed three distinct clusters associated with each molecular subtype (Supplementary Fig. 10a). We created three subtype-

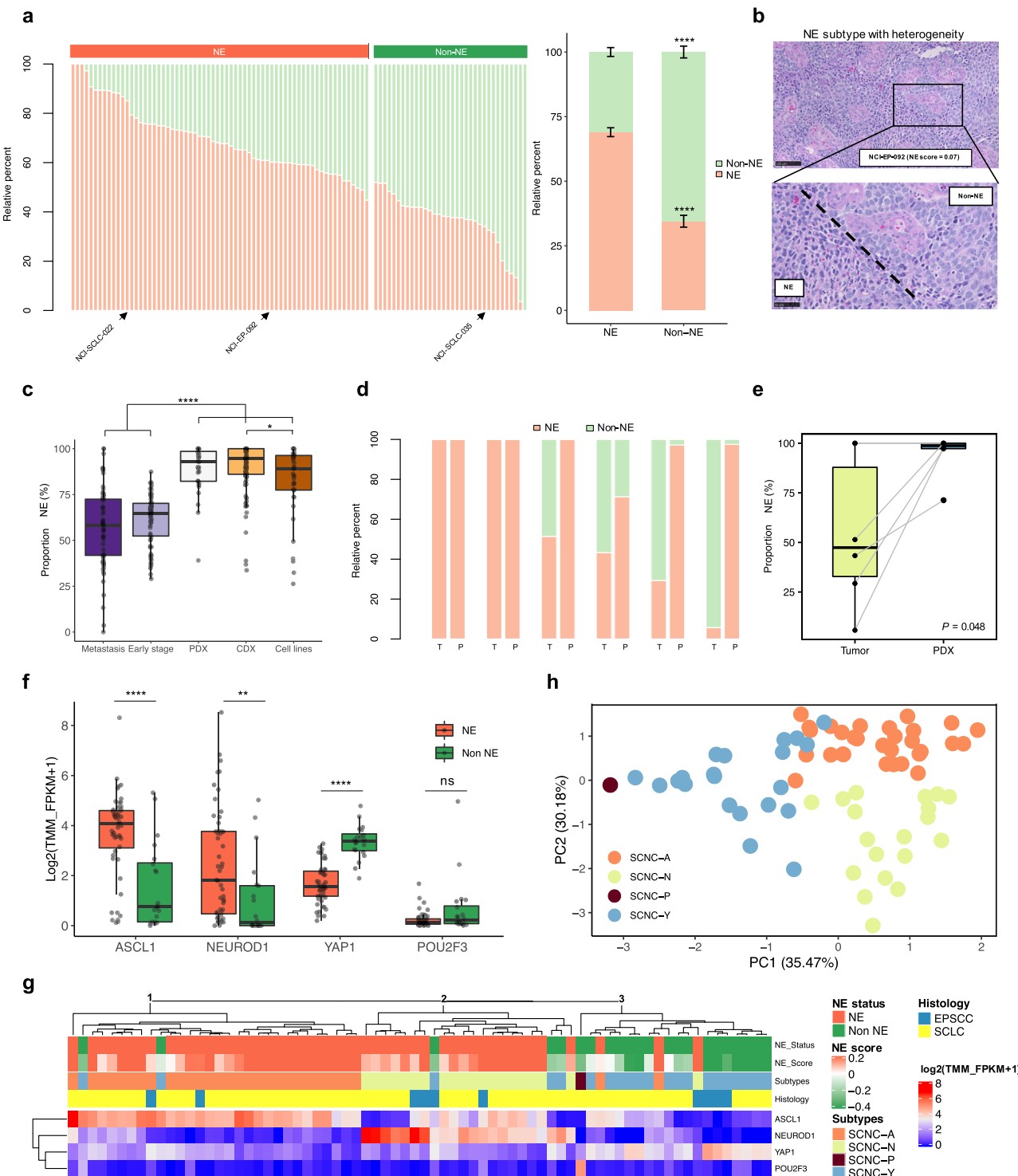

specific gene signatures from the top 500 contributors to the first and second principal components (Supplementary Data 1). Interestingly, *NEUROD1* and *ASCL1* were among the top contributing genes to PC2. *YAP1* was not predicted to contribute as strongly to PC1. However, another core component of the Hippo signaling pathway, *LATS2* kinase, which directly phosphorylates YAP1, and *TGFBR2*, whose downstream signaling was shown to interact with the Hippo signaling pathway[40,41], were among the top 10 negative contributors to PC1. The unsupervised analysis of the subtype-specific gene lists revealed three main clusters, consistent with the expression of the three transcription

factors (Supplementary Fig. 10b). We also observed overlap with previously published gene sets[27,31,35,42], further supporting the subtype-defining transcriptional signatures. Collectively, these results demonstrate robust stratification of metastatic SCNCs by the expression of the lineage-defining transcription factors.

**Transcriptional programs associated with neuroendocrine differentiation**. Transcription factors function as molecular switches to regulate the expression of cell-type or lineage-specific target genes. We investigated whether the expression of the subtype-defining transcription factors correlated with specific

**Fig. 2 Intratumoral heterogeneity of neuroendocrine differentiation in patient tumors and model systems. a** CIBERSORT analysis[25] of the 50-gene signature in 100 tumors grouped by NE subtype (left stacked bar chart). Relative proportion of NE and non-NE cells within each SCNC NE subtype (right box plot) ($n = 100$ tumors; data are presented as mean ± SEM). Two-tailed Student-$t$ test, ****$P = 2.07$e-20. **b** Representative photomicrograph images of H&E-stained small cell cancer of the NE subtype with heterogenous morphological features (observations were repeated independently two times). **c** Intratumoral proportion of NE cells based on CIBERSORT deconvolution in 84 recurrent and metastatic tumors from the current cohort and previously described cohorts of 81 early-stage tumors[30], 32 PDX, 120 CDX[22,28] models, and 39 immortalized cell lines[33,34]. Kruskal–Wallis test followed by Dunn's multiple comparisons test with BH correction, ****$P < 0.0001$ (ranging from $P = 2.13$e-26 to $2.08$e-07), *$P = 0.023$. **d** CIBERSORT analysis[25] of the 50-gene signature in six patient-matched tumor biopsies (T) and xenograft tumors (P). **e** Proportion of NE cells based on CIBERSORT deconvolution in 6 PDX and corresponding donor patient tumors. Paired-$t$-test, $P = 0.048$ (**f**) Box plots showing mRNA levels in NE and non-NE tumors for the four transcription factors. Two-tailed Mann–Whitney U-test, ****$P < 0.0001$ (ranging from $P = 2.53$e-09 to $4.56$e-05), **$P = 0.00103$, ns, not significant ($n = 72$ patients). **g** Heatmap generated by unsupervised hierarchal clustering of the four transcription factors in 72 patients. Neuroendocrine scores and NE status derived from the 50-gene signature, and the molecular subtypes derived from the clustering and the histology are indicated above the heatmap. **h** Supervised PCA using the expression of the four transcription factors. Each dot represents a patient colored by the transcriptomic category. All tests are two-tailed. All box plots indicate the inter-quartile range (IQR), the middle line corresponds to the median, and the upper and lower whiskers represent observations within 1.5*IQR (Q3 + 1.5*IQR or Q1 − 1.5*IQR). Abbreviations: NE neuroendocrine differentiation; TMM Trimmed Mean of M-values; FPKM Fragments Per Kilobase of Exon Per Million Fragments Mapped; PCA principal component analysis. PDX patient-delivered xenografts; CDX CTC-derived xenografts.

transcription profiles, contributing to inter- and intratumor heterogeneity. *MYC* paralogs are frequently activated in SCLC, with individual paralogs overexpressed in a mutually exclusive manner[30]. As a target of *ASCL1*, *MYCL* is highly expressed in SCLC-A and necessary for its development[35]. *MYC* on the other hand is a target of *NEUROD1* and drives a non-NE phenotype[22,35,43]. We found that *MYCL* expression was upregulated in NE tumors of SCNC-A, whereas higher levels of *MYC* was observed in non-NE tumors of SCNC-Y (Fig. 3a, b). There were no differences in *MYCN* expression between NE and non-NE tumors, but an elevated expression was noted specifically in SCNC-N tumors.

Given the recently discovered role of MYC in driving the temporal evolution of SCLC from NE to non-NE fate[31,44], we sought to estimate the intratumor heterogeneity in terms of MYC-driven tumor progression. Using gene signatures expressed during the course of NE to non-NE tumor transition[31], tumors were predicted to have cells at different stages of MYC-driven progression (Fig. 3c). SCNC-A and SCNC-N subtypes harbored a significantly higher proportion of the NE-high early time point signature, whereas the non-NE late time point signature was significantly enriched in the MYC-high, SCNC-Y subtype (Fig. 3d). Analysis of serial sections revealed similar time-course gene expression signatures among tumors, indicative of intratumoral homogeneity at a given time. Sequential biopsies and biopsies from distinct sites exhibited more variations in the proportion of cells in transition (Supplementary Fig. 10c). Interestingly, in comparison to early-stage, treatment-naïve SCLCs[30], the metastatic and relapsed tumors in this cohort harbored lower proportion of the mid-late time point signature and a significantly higher proportion of the late time point signature (Supplementary Fig. 10d).

The MYC-driven temporal shift of SCLC from NE to non-NE state is promoted by activation of Notch signaling[24,31]. Several Notch pathway genes identified as MYC targets were differentially regulated, consistent with MYC expression in tumors[31] (Supplementary Fig. 11a). NOTCH transcripts (NOTCH1, −2, −3) and the NOTCH target RE1 silencing transcription factor (REST) were downregulated in the NE subtypes (Fig. 3e, f). In contrast, NOTCH inhibitory ligands *DLL1* and *DLL3* were significantly upregulated in NE tumors of the SCNC-A subtype, consistent with them being targets of ASCL1[24,45]. Consistent with Notch signaling promoting epithelial-mesenchymal transition (EMT)[46], non-NE tumors demonstrated higher expression of mesenchymal marker vimentin (VIM) and transforming growth factor beta (TGF-β), an inducer of EMT and negative regulator of *ASCL1*[47,48]. By contrast, the epithelial markers E-cadherin

(CDH2) and EpCAM were upregulated in NE tumors (Supplementary Fig. 11b), and a higher EMT signature score[49] was associated with lower neuroendocrine scores (Fig. 3g). Of note, we observed enrichment of two cancer-associated fibroblast (CAF) subpopulation-specific signatures[50] (inflammatory, iCAFs and myofibroblastic, myCAFs) in non-NE tumors, consistent with the role of CAFs in promoting EMT in cancer cells[51] (Supplementary Fig. 11c).

To further confirm the association between SCLC molecular subtypes and each gene significantly differentially expressed between NE and non-NE tumors, we performed a PCA (Fig. 3h). In agreement with our previous results, the PCA showed convergent expression of *ASCL1* with *MYCL, DLL1, -3, CDH2* and *EPCAM*, *NEUROD1* with *MYCN*, and *YAP1/POU2F3* with *MYC, NOTCH1, -2, -3, REST, VIM,* and *TGFB1*. Together, these analyses demonstrate heterogeneity within individual tumors, with each tumor harboring cells of more than one SCLC subtype and various developmental stages, while being driven by a dominant transcriptional program.

**Biological features associated with neuroendocrine differentiation.** Next, we sought to understand the key biological features of the SCLC subtypes that may in turn reveal subtype-specific vulnerabilities. DNA replication stress is recognized as a SCLC hallmark and nearly all the active chemotherapeutics in SCLC are DNA damaging agents[52]. NE relative to non-NE tumors exhibited marked upregulation of genes essential for cancer cells to cope with the increased rates of replication (*TOP2A, MCM3*), and prevent replication stress-induced DNA damage, including those related to the DNA damage response (*CHEK1, -2, TP53BP1, TOPBP1*), cell cycle progression (*CDC25A, -B, -C, CDK1, -2, AURKA, -B, CDK1, -2, PLK1, CCNB1, -2, CCNA2*), and DNA repair (*PARP1, -2, BRCA1, RAD51, PRKDC*) (Fig. 4a, S11d). Accordingly, signatures of replication stress response[53] and cell cycle[54], correlated positively with neuroendocrine score (Fig. 4b, Supplementary Fig. 12a). *EZH2* which encodes a histone methyltransferase that promotes chromatin compaction and transcriptional silencing was highly expressed in NE tumors. Previous studies have noted the role of *EZH2* in acquired resistance to chemotherapy via *SLFN11* silencing[55,56]. We found no difference in *SLFN11* expression between NE and non-NE tumors (Supplementary Fig. 11d).

Immune checkpoint inhibitors (ICI) are now part of standard care of patients with SCLC, with benefit observed in a small subset[57–59]. Evaluation of the immune contexture identified greater expression of several immune-related genes in non-NE tumors, including genes associated with antigen presentation (*B2M, HLA* genes), IFN-γ signaling (*IFIT1, -2, -3, IFITM*),

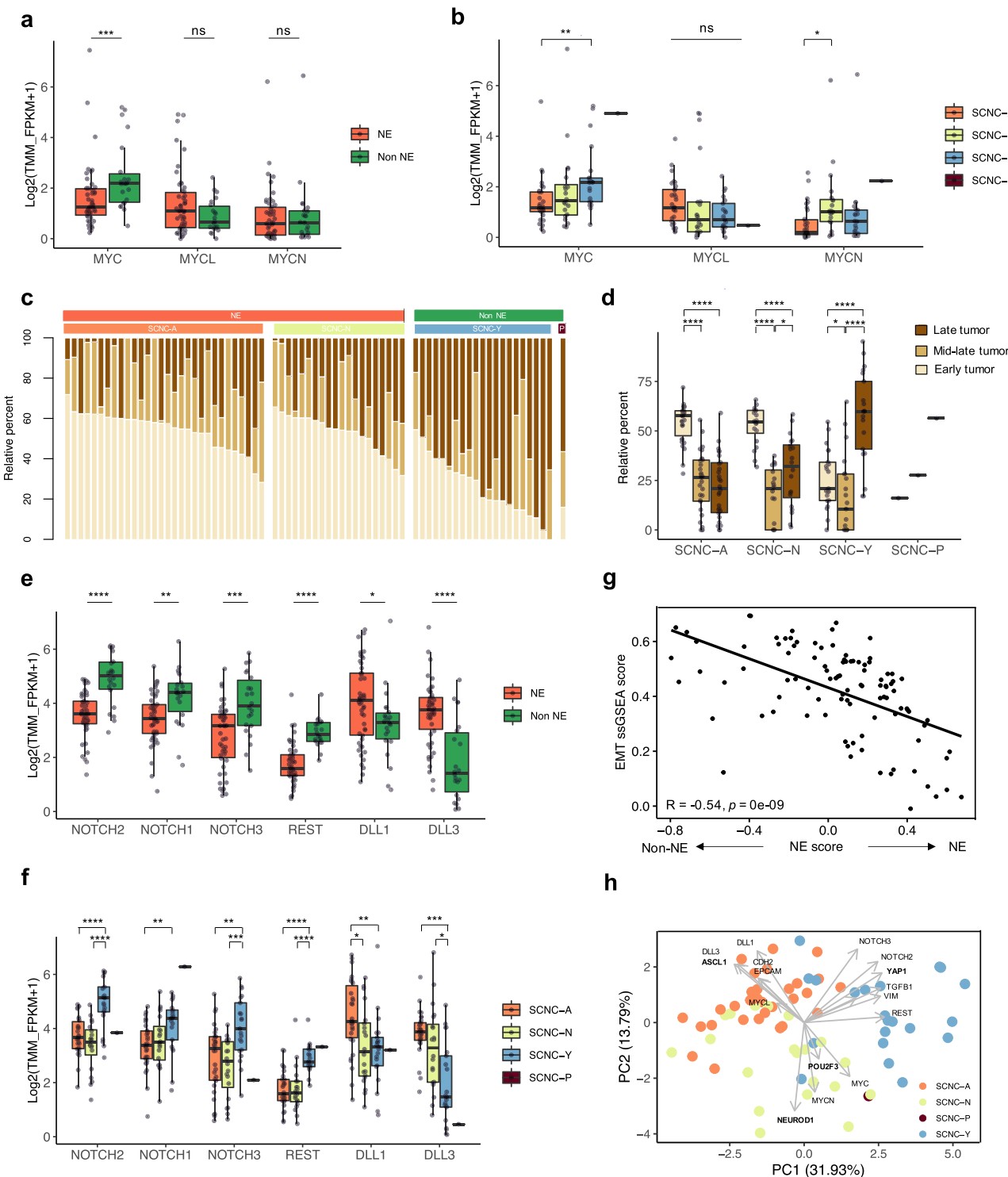

cytotoxic T-cell activity and adaptive immune resistance (*CD8A, -B, CD4, CD274*) (Fig. 4a, Supplementary Fig. S11e). Correspondingly, gene signatures related to immune cell infiltration[60,61], IFN-γ signaling[62], and antigen presentation[63] were negatively correlated with neuroendocrine score (Fig. 4c, Supplementary Fig. 12a). A comprehensive analysis of tumor infiltrating immune cells using ssGSEA[60] and CIBERSORT[25,64] revealed enrichment of several immune cell populations in non-NE relative to NE tumors[25], including monocytes, macrophages, natural killer (NK) cells, T cells and dendritic cells (Supplementary Fig. 12a–c), consistent with

previous studies[65,66]. The higher T-cell infiltration was confirmed by CD3 IHC staining (Supplementary Fig. 12d).

To gain insights into biological pathways underlying the neuroendocrine subtypes, we examined differentially enriched genesets between NE and non-NE SCNC (Fig. 4d). Genes related to cell cycle, proliferation, and DNA repair were enriched in NE tumors, indicative of these tumors harboring a replication stress phenotype, while immune response, metabolism and cell adhesion genesets were enriched in non-NE tumors (Fig. 4e). Consistently, gene ontology (GO) analysis of the subtype-specific

**Fig. 3 SCNC subtypes exhibit unique transcriptional programs. a, b, e, f** Box plots showing mRNA levels in NE and non-NE tumors or in SCNC molecular subtypes, for (**a**, **b**) the *MYC* family of genes and (**e**, **f**) the Notch signaling pathway. Two-tailed Mann–Whitney *U*-test, ****$P < 0.0001$, ***$P < 0.001$, **$P < 0.01$, *$P < 0.05$, ns, not significant ($n = 72$ patients). **c** CIBERSORT analysis[25] of the gene signatures derived at different time points of MYC-driven tumor transition toward a non-NE phenotype[31], in 72 patients grouped by molecular subtype. **d** Box plot of the relative proportion of early, mid/late and late tumor phenotypes within each SCNC molecular subtype. Two-tailed Mann–Whitney *U*-test with BH adjustment, ****$P < 0.0001$, *$P < 0.05$ ($n = 72$ patients). **g** Pearson correlation between the 50-gene signature score and an epithelial-mesenchymal transition (EMT) score determined by ssGSEA[49]. Pearson's R value and *P*-value are indicated. **h** Supervised PCA of gene expression data for 12 selected genes associating with SCNC transcriptome subtype, MYC, and Notch signaling. Each dot represents a patient colored by the transcriptomic category. Gray arrows correspond to PCA loadings. All tests are two-tailed. All box plots indicate the inter-quartile range (IQR), the middle line corresponds to the median, and the upper and lower whiskers represent observations within 1.5*IQR (Q3 + 1.5*IQR or Q1 − 1.5*IQR). Abbreviations: NE neuroendocrine differentiation; TMM Trimmed Mean of M-values; FPKM Fragments Per Kilobase of Exon Per Million Fragments Mapped; PCA principal component analysis; EMT epithelial-mesenchymal transition; ssGSEA single sample gene set enrichment analysis.

transcriptional signatures revealed enrichment of DNA damage response and cell cycle related genes in SCNC-N tumors, and inflammation and immune response related genes in SCNC-Y tumors (Supplementary Data 2). The most significantly enriched metabolic pathways in non-NE tumors included lipid metabolism, nicotinate, and nicotinamide metabolism (Supplementary Data 3). The KEGG arginine and proline metabolic pathway, whose role in MYC-driven SCLC was recently reported[67], was also significantly enriched in non-NE tumors (Supplementary Fig. 12e). MYC was also shown to regulate purine biosynthesis and ribosome biogenesis in SCLC[68,69], however, the associated KEGG pathways were similarly enriched in both tumor subtypes.

This analysis also revealed a subset of tumors (11/72, 15.3%) with hybrid NE and non-NE phenotypes, remarkable for enrichment of xenobiotic metabolism and drug transporter genesets (Fig. 4d, f), reminiscent of the neuroendocrine variant (NEv2, also termed SCLC-A2) subtype described in SCLC cell lines[42]. NEv2-like tumors were more likely to be derived from liver metastases (9/11, 81.8%), and NEv2-like score was significantly higher in liver metastases compared with other biopsy sites (Supplementary Fig. 12f). To exclude the possible confounding influence of biopsy site, we compared tumors from liver biopsies and other metastatic sites. Among the 90 pathways specifically upregulated in liver-derived tumors, none were overlapping with NEv2-like pathways (Supplementary Table 6). Collectively, these analyses define biological processes unique to NE and non-NE SCNC. The NEv2-like subset of tumors with mixed NE and non-NE gene expression may represent a transitional, high-plasticity cell state[70], that may portend drug resistance and poor prognosis.

**Genomic alterations associated with neuroendocrine differentiation.** We next sought to evaluate the genomic alterations across NE and non-NE tumors. WES was performed on 34 SCNCs including 27 SCLCs and 7 EPSCCs; Supplementary Fig. 1). Tumor mutation burden (TMB) was significantly higher in NE tumors (Fig. 5a). Consistent with the major role of tobacco mutagenesis in SCLC most tumors harbored smoking signatures (SB4, 5 and 29)[71]. There were no significant differences in mutational signatures between NE and non-NE tumors (Fig. 5b). Somatic alterations and/or copy number loss of *TP53* and *RB1* were frequently observed (*TP53* 27/34, 79.4%; *RB1* 26/34, 76.5%) (Fig. 5c), with a higher frequency of *RB1* alterations in NE compared with non-NE tumors (20/22, 90.9% vs. 6/12, 50.3%, $P = 0.013$ by Fisher's exact test), as described previously[36]. Consistent with loss of *NOTCH* function promoting neuroendocrine differentiation, *NOTCH1* mutations were observed only in NE tumors (4/22, 18.2% NE vs. 0/12, 0% non-NE). Chromatin modifier genes were more frequently altered in NE compared with non-NE tumors (14/22, 63.6% vs. 1/12, 8.3%, in

NE vs. non-NE, respectively, $P = 0.003$ by Fisher's exact test). Gain or amplification of *MYCL* was also more frequent in NE tumors (13/22, 54.5% vs. 1/12, 8.3%, $P = 0.0011$ by Fisher's exact test). Together these analyses reveal relative genomic homogeneity between neuroendocrine subtypes, with notable exceptions of *RB1*, *NOTCH*, *MYCL*, and chromatin modifiers.

**Treatment responses and survival of NE subtypes.** Although several studies have suggested that SCLC subtypes may have unique therapeutic vulnerabilities, these are largely based on preclinical models and investigated dependencies to a single gene, pathway, or drug[37,43,67,68,72–74]. We leveraged the transcriptomic data and detailed clinical annotations to study potential subtype-specific therapeutic vulnerabilities. All patients received platinum-based chemotherapy as their first line treatment. Most patients in our cohort (64/72, 88.9%) were treated at relapse with ICI in monotherapy or in combination with a poly (ADP-ribose) polymerase (PARP) inhibitor[12,14], or ataxia telangiectasia and rad3 related (ATR) inhibitor plus topotecan[11,13]. Five patients received both combinations sequentially. Only patients (60/72, 83.3%) with tissue sampling prior to study treatment were included in the subgroup analysis.

There were no significant differences in the clinical characteristics (Supplementary Table 1) or survival between patients with NE and non-NE tumors (median OS, 13.7 vs. 14.3 months; HR, 1.28; 95% CI, 0.75–2.17) (Fig. 6a). Patients treated with ICI had significantly shorter PFS than patients receiving ATR inhibitor (median PFS, 1.7 vs. 2.8 months; HR, 1.85; 95% CI, 1.06–3.23, log-rank $p$-value = 0.03) (Supplementary Fig. 13a, b). Among ICI-treated patients, those with non-NE tumors derived greater clinical benefit compared to patients with NE tumors (30.8% vs. 7.1%; Chi-square, $P = 0.046$) (Fig. 6b). In agreement with recent observations linking Notch activation, non-NE differentiation and tumor immunity[14], tumors of patients with clinical benefit from immunotherapy tended to have higher expression of Notch pathway genes than those without clinical benefit (Supplementary Fig. 13c). Patients with NE tumors had higher overall response and clinical benefit when treated with ATR inhibitor compared to immunotherapy (ORR: 33.3% vs. 7.1%; Chi-square, $P = 0.027$ and clinical benefit rate: 46.7% vs. 7.1%; Chi-square, $P = 0.0024$) (Fig. 6b, Supplementary Fig. 13d). Patients with NE tumors had significantly shorter PFS and a trend towards shorter OS when treated with immunotherapy compared with ATR inhibition (median PFS, 1.5 vs. 2.8 months; HR, 2.37; 95% CI, 1.19–4.70; log-rank $p$-value = 0.012, Fig. 6c; median OS, 2.9 vs. 6.2 months; HR, 1.65; 95% CI, 0.85–3.23; log-rank $p$-value = 0.14, Supplementary Fig. 13e). Multivariate analysis revealed investigational therapy as the only variable significantly associated with PFS (Fig. 6d). Patients with NE tumors were more likely to progress on immunotherapy than with ATR inhibition after adjusting for age, sex, smoking, stage at diagnosis, platinum sensitivity and

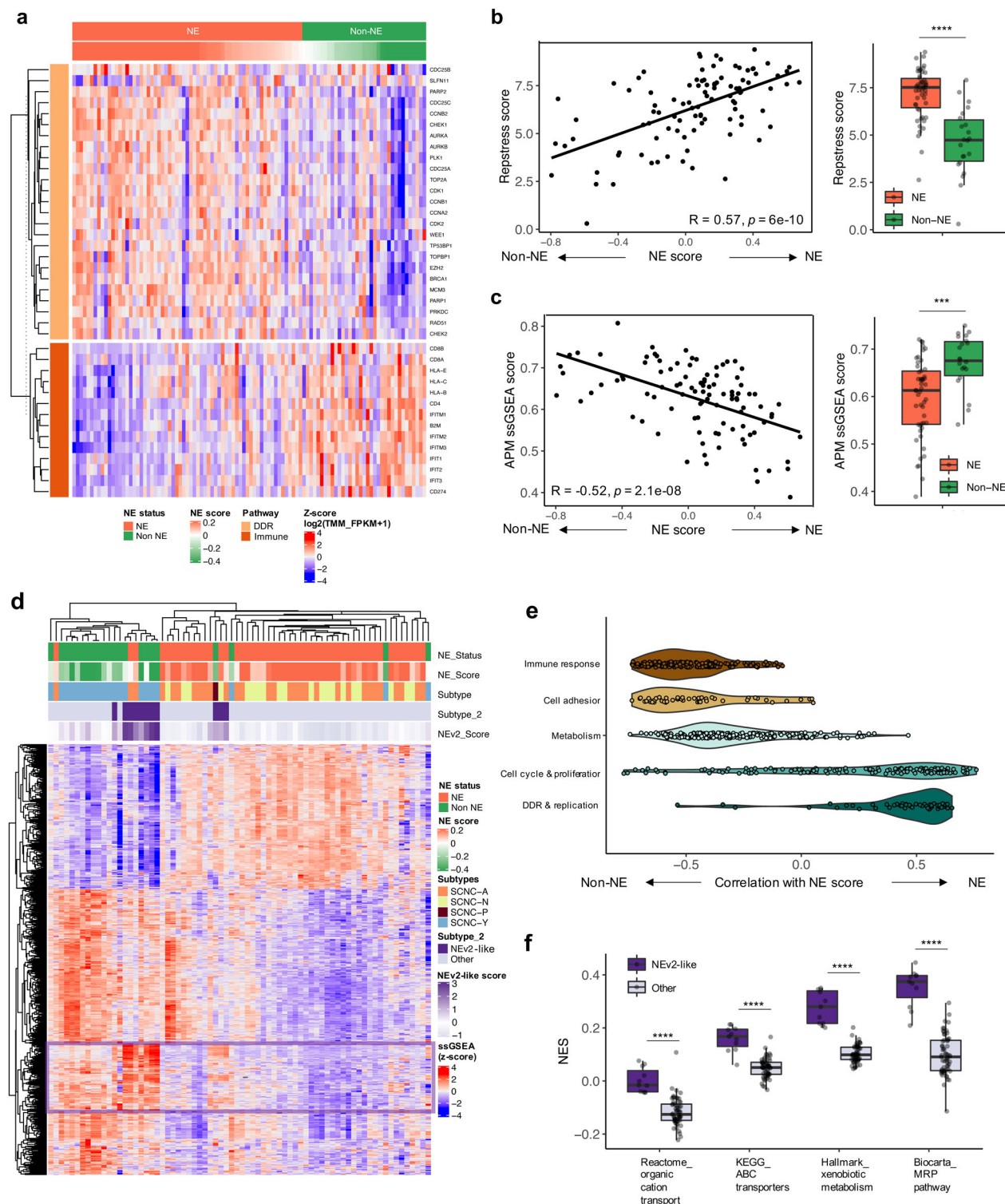

number of systemic therapies (HR, 2.87; 95% CI, 1.1–7.5; $P = 0.031$).

Importantly, patients with NEv2-like tumors had significantly poorer OS compared with the rest of the cohort (9.8 vs. 15.0 months; HR, 2.45; 95% CI, 1.25–4.79; log-rank $p$-value = 0.0072) (Fig. 6e). The association remained significant after adjusting for co-factors including age and stage at diagnosis, sex, and platinum sensitivity (HR, 2.32; 95% CI, 1.16–4.65; $P = 0.018$) (Supplementary Fig. 13f). Consistently, compared with the other subtypes, the NEv2-like subtype was more likely to have a higher

proportion of platinum-resistant tumors (9/11, 81.8% vs. 35/61, 57.4%) and liver metastasis (11/11, 100% vs. 16/61, 26.2%), an independent predictor of poor prognosis in SCLC (Supplementary Fig. 13g, h). Furthermore, none of the five patients with NEv2-like subtype tumors responded to ATR inhibition (Supplementary Fig. 13i), and these patients had shorter PFS (2.3 vs 3.8 months; HR, 2.69; 95% CI, 1.77–9.39; Supplementary Fig. 13j) and significantly shorter OS (2.7 vs 9.6 months; HR, 3.62; 95% CI, 1.02–12.78; log-rank $p$-value = 0.033; Supplementary Fig. 13k) than patients with other subtypes treated with ATR inhibition.

**Fig. 4 SCNC subtypes are characterized by distinct biological features. a** Heatmap of 25 DDR and 14 immune genes in 100 tumors. Neuroendocrine scores and NE status derived from the 50-gene signature are indicated on top. **b, c** Pearson correlation between the 50-gene signature score and (**b**) a replication stress response score[53] or (**c**) an antigen processing and presenting machinery (APM) signature score[63]. Pearson's R values and *P*-values are indicated. A box plot of the distribution of the signature scores between NE and non-NE tumors is shown on the right of each graph. Two-tailed Mann–Whitney *U*-test, ****$P$ = 3.79e-07, ***$P$ = 0.0004 ($n$ = 72 patients). **d** Heatmap clustered with Pearson's correlation and average linkage of the top 1000 pathways differentially regulated between NE and non-NE tumors. The NEv2-related pathways are highlighted by a purple square on the heatmap (**e**) Distribution of correlations between neuroendocrine scores and selected pathway gene sets from BioCarta, Hallmark, KEGG, PID and Reactome. **f** Box plot of the NES for four xenobiotic metabolism and drug transporter pathways in NEv2-like and other tumors. Two-tailed Mann–Whitney *U*-test, ****$P$ < 0.0001 (ranging from $P$ = 1.72e-07 to 1.43e-06) ($n$ = 72 patients). All tests are two-tailed. All box plots indicate the inter-quartile range (IQR), the middle line corresponds to the median, and the upper and lower whiskers represent observations within 1.5*IQR (Q3 + 1.5*IQR or Q1 − 1.5*IQR). Abbreviations: NE Neuroendocrine differentiation; TMM Trimmed Mean of M-values; FPKM Fragments Per Kilobase of Exon Per Million Fragments Mapped; DDR DNA damage response; ssGSEA single sample gene set enrichment analysis; NES Normalized Enrichment score.

These findings suggest that differential treatment responses in SCNC subtypes and warrants prospective testing in well-defined cohorts.

## Discussion

SCLC is an exceptionally lethal malignancy, which to date is treated as a homogenous disease with identical treatments for all patients. While SCLC molecular subtypes defined by neuroendocrine differentiation have been described in human primary tumors and cell lines, the biological features or therapeutic vulnerabilities of metastatic SCLC are not known. Here, we present an integrative analysis of histology, expression profiles, genetic alterations, and outcomes of metastatic SCLC, representing the spectrum of neuroendocrine differentiation. Our findings from the largest such cohort reported to date: (i) confirm transcriptional subtypes driven by neuroendocrine differentiation in metastatic and recurrent SCLCs, (ii) reveal remarkable intra-tumor heterogeneity of neuroendocrine differentiation in metastatic SCLCs, notably not recapitulated in patient-derived model systems, (iii) identify a clinical subtype with hybrid NE and non-NE phenotypes defined by drug resistance and particularly poor outcomes, and (iv) reveal potential subtype-specific vulnerabilities related to high replication stress and Notch-driven immune activation, respectively in NE and non-NE tumors. These findings (Fig. 7) have implications for rationally targeted treatment approaches and may ultimately help improve clinical outcomes for patients with this historically recalcitrant cancer.

Metastatic tumors from our cohort exhibited co-expression of NE and non-NE gene programs at varying levels in nearly all cases, with a higher proportion of tumors exhibiting co-expression of ASCL1 and NEUROD1 than early-stage tumors and preclinical models[4,30]. Studies in preclinical models and relapsed SCLC samples show transcriptional flexibility resulting in higher non-NE differentiation in more advanced and treatment-resistant tumors[31,32]. A higher frequency of non-NE tumors were observed in our cohort of metastatic and relapsed tumors than previously reported early-stage, treatment naïve tumors[4,27,38], supporting the notion that therapy selects for and promotes dynamic evolution of SCLCs to a more non-NE phenotype[21,27,31,32]. Non-NE tumors of SCNC-P subtype were however less frequent, likely due to their exceptionally poor prognosis[27]. The role of YAP1 as a subtype-defining marker remains unclear. Although our results are in agreement with recent work reporting YAP1 expression in a subset of SCLCs[75], other studies suggest that YAP1 does not define a distinct subgroup[27,38]. Serial sampling of larger cohorts over their treatment course and further dissection of intratumoral heterogeneity using single-cell RNA-seq (scRNA-seq) will be needed to clarify the plasticity between subtypes.

We also observed intertumoral heterogeneity at the genomic level with inactivating NOTCH1 and RB1 mutations, and MYCL

amplifications occurring at higher frequency in the NE subtype tumors[31,36]. Importantly, NE tumors were significantly more likely to harbor mutations in chromatin modifier genes, supporting the role of epigenetic dysregulation in the development of NE features.

While PDXs are documented to largely recapitulate the polygenomic architecture of human tumors[29,76], direct comparisons of molecular characteristics between patient tumors and model systems are limited. We found that patient-derived models are notably enriched for NE programs than human SCLC, an observation that was confirmed in an independent cohort of patient-matched tumors and PDXs. The lack of SCLC transcriptional diversity in model systems may be due to clonal selection for a dominant transcription factor under experimental conditions or represent tumor evolution in the absence of TME components. The positive correlation between tumor purity and neuroendocrine score supports this last hypothesis. These findings have potential implications for clinical translation of PDX therapeutic responses[77].

The prognostic and predictive significance of neuroendocrine subtype classification is not well defined. Non-NE SCLC defined by lack of INSM1 expression was associated with increased chemoresistance and a trend towards shorter patient survival in a previous report[36]. We found no significant difference in overall survival between the SCLC subtypes. However, exploration of the predictive significance of neuroendocrine subtype classification for investigational therapies revealed that NE tumors preferentially respond to replication stress targeted therapies, while ICI-treated patients with non-NE tumors derived greater benefit compared to those with NE tumors. The subtype-specific vulnerabilities hypothesized here need to be tested in prospective clinical trials. Finally, we identified a subset of SCNCs characterized by enrichment of xenobiotic and drug transporter pathways, and exceptionally poor response to treatment and survival. Interestingly, the NEv2-like subtype was more frequently identified in liver biopsies. Previous studies have shown that liver metastasis is an independent predictive factor of poor survival in SCLC[78–80], in line with the high aggressiveness of the NEv2-like subtype. Further studies are warranted to evaluate core biological characteristics and therapeutic targets in this chemo-resistant subtype.

Our study has several limitations. First, due to the limited tissue availability, the intratumoral heterogeneity, the TME composition and the EMT were inferred computationally through ssGSEA and CIBERSORT deconvolution of bulk RNA-seq data. Although our results are generally in agreement with findings from scRNA-seq[31,66] and SCLC cell lines[33], which lack the TME, they remain to be validated by unbiased approaches. Secondly, the outcomes of the ICI-treated cohorts may have been confounded by combination therapy with PARP inhibitor, but multiple studies suggest no added benefit to combining PARP inhibitors with ICI[12,77]. Finally, subtype-specific

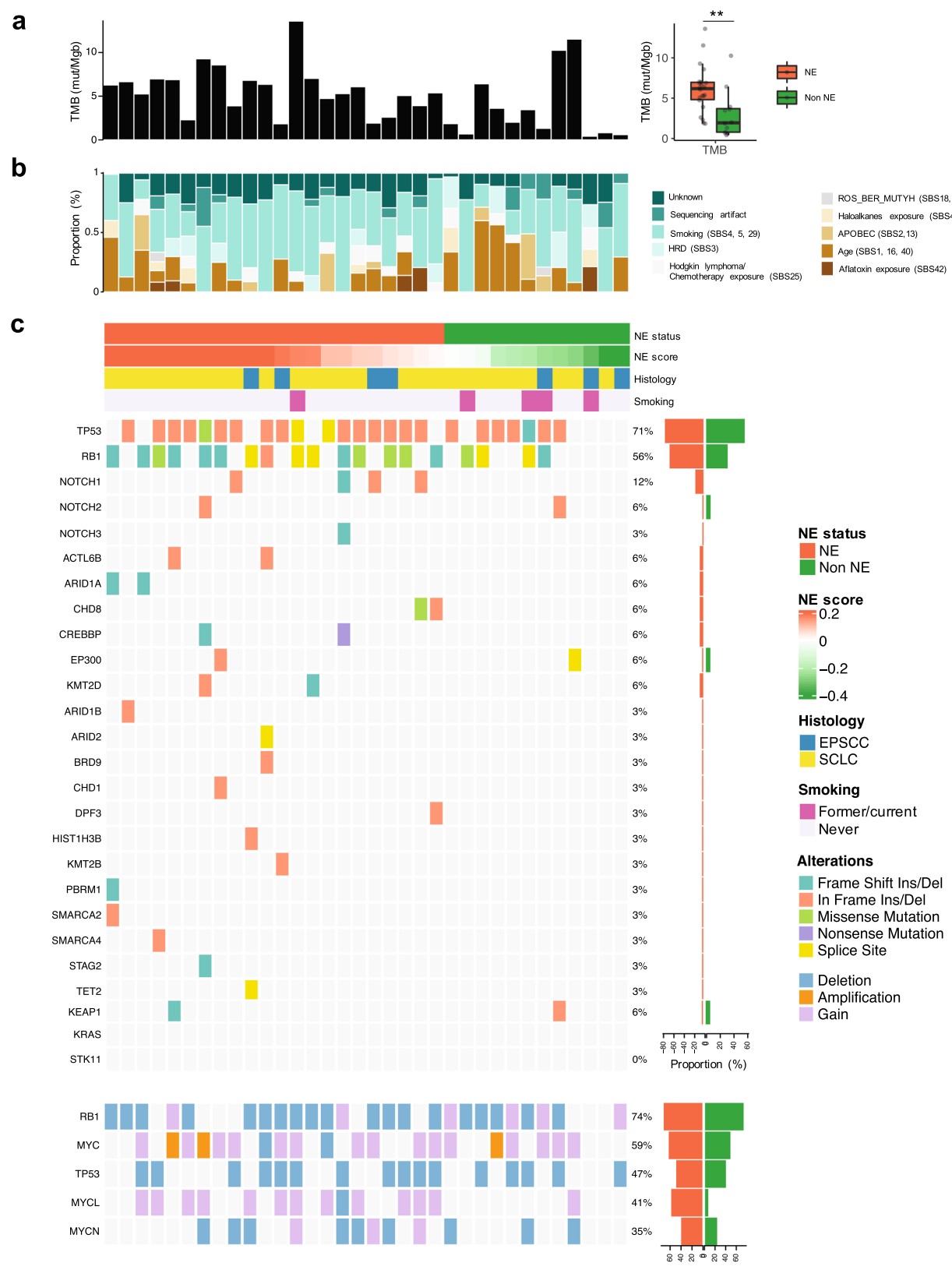

treatment vulnerabilities will require further validation in a prospective study.

In conclusion, this study provides comprehensive analyses of transcriptomic and genetic heterogeneity in metastatic SCLC, along with detailed clinical annotation and outcomes, illustrating the genetic and transcriptional complexity in SCLC patient tumors, and provide a rational framework for prospectively targeting the intertumoral heterogeneity.

## Methods

**Study design**. We undertook a retrospective study of genomic and clinical characteristics in transcriptomically defined NE differentiated SCNC samples from patients

**Fig. 5 Genomic alteration profiles of SCNC subtypes.** Genomic characteristics of NE ($n = 22$) and non-NE ($n = 12$) tumors. **a** Bar graph of TMB in 34 patients. Two-tailed Mann–Whitney $U$-test, **$P = 0.0021$. **b** Mutational signature proportions[71]. **c** Neuroendocrine status (NE vs. non-NE), 50-gene signature score, histology (SCLC vs. EPSCC) and smoking status (former/current vs. never). Top heatmap indicates mutations in genes of *TP53*, *RB1*, *NOTCH* paralogues, and chromatin modifiers[90]. Bar charts on the right of the heatmap indicates frequency of mutations in NE vs. non-NE tumors. Bottom heatmap shows copy number alteration of *TP53*, *RB1*, and MYC paralogue genes. Box plots indicate the inter-quartile range (IQR), the middle line corresponds to the median, and the upper and lower whiskers represent observations within 1.5*IQR (Q3 + 1.5*IQR or Q1 − 1.5*IQR). Abbreviations: NE neuroendocrine differentiation; TMB total mutational burden; COSMIC The Catalog of Somatic Mutations in Cancer; SCLC small cell lung cancer; EPSCC extrapulmonary small cell cancer.

who received care at the National Cancer evaluated the contributions of transcriptomic NE differentiation subtype-specific therapeutic vulnerabilities to combinations of (1) ATR inhibitor berzosertib and topoisomerase inhibitor topotecan (ClinicalTrials.gov # NCT02487095; NCI protocol #15-C-0150) ($n = 27$ patients); and (2) immune checkpoint inhibitor durvalumab and PARP inhibitor olaparib (ClinicalTrials.gov # NCT02484404; NCI protocol #15-C-0145) ($n = 18$ patients) or immune checkpoint inhibitor alone ($n = 25$ patients). Details of the two clinical studies were previously reported[11–14]. We also sequenced tumor samples collected from SCNC patients who were enrolled in the NCI thoracic malignancies natural history protocol (Clinical-Trials.gov # NCT02146170; NCI protocol #14-C-0105). NIH IRB, Office of Human Subjects Research Protections at NCI approved the studies; all patients provided written informed consent for tumor and matched normal sample sequencing.

**SCLC patient derived xenograft model.** Eight-week-old male and female NSG mice (NOD.Cg-Prkdc scid Il2rg tm1Wjl/SzJ; # 005557, The Jackson Laboratory, Bar Harbor, ME) were implanted subcutaneously with fresh patient-needle biopsy supported with Matrigel (Corning). For each PDX, one mouse was used to start the expansion and then two mice at each passage. Consistent $7 \times 7$ mm3 were implanted from passage P1 to P3 while maintaining tumor stock at each passage. Mice were monitored daily, with caliper measurements and body weights recorded bi-weekly; caliper monitoring was performed three times per week if necessary, for close monitoring. The patient derived xenograft (PDX) model was well characterized for consistency and reliability in vivo and at the histopathological level for small cell lung cancer and similarity with patient histopathology of origin.

The PDX tumor stock are viably frozen to ensure early passage of the model are well preserved. For any study plan with the model, the SCLC-PDX is revived through tumor passage in increasing number of recipient mice; for quality control the histopathology of donor tumors is verified at every passage as well as the take up rate. Before implanting any preclinical study cohort, we ensure that take up rate is at 100% in 2 donor passages prior study cohort implant. Mouse handling and procedures were conducted under an approved Animal Study Protocol according to Frederick National Laboratory Animal Care and Use Committee guidelines. Mice were on 12 h light/dark cycle. Temperature of the rooms was between 68–74F, and humidity range was 30–70%.

**DNA and RNA sequencing.** FFPE tumor tissue samples or frozen tumor samples in selected samples were prepared for RNA-Seq and WES. One hundred nanograms of DNA was sheared to approximately 200 base pairs (bp) by sonication (Covaris, Woburn, MA). Exome enrichment was performed using SureSelect Clinical Research Exome Kits according to the manufacturer's instructions (Agilent, Santa Clara, CA) and RNA enrichment was performed using TruSeq RNA Exome Library Prep according to manufacturer's instructions (Illumina, San Diego). Paired-end sequencing ($2 \times 75$ bp) was performed on an Illumina Next-Seq500 instrument. The sequences were compared to the human reference genome hg19 using internally developed ClinOmics somatic Bioinformatic Pipeline v3.1. Peripheral blood DNA extracted from individual patients was used for germline exome sequencing. In brief, raw sequencing data in FASTQ format were aligned against the reference human genome (hg19) with BWA. The Genome Analysis Toolkit (GATK) and HaplotypeCaller (HAPLOC) were used for germline SNV and indel calling; whereas Strelka was used for somatic single nucleotide variant (SNV) and small indel calling. ANNOVAR was used to functionally annotate genetic variants. Variants with variant allele frequency (VAF) > 0.10, tumor sequencing depth > 50, and matched germline sequencing depth > 50 were considered. FACETS algorithm was used to determine total and allele-specific DNA copy number from WES. RNA was extracted from FFPE tumor cores using RNeasy FFPE kits according to the manufacturer's protocol (QIAGEN, Germantown, MD). RNA-seq libraries were generated using TruSeq RNA Access Library Prep Kits (TruSeq RNA Exome kits; Illumina) and sequenced on NextSeq500 sequencers using 75 bp paired-end sequencing method (Illumina, San Diego, CA). For transcriptomic analyses, raw RNA-Seq count data were normalized for inter-gene/sample comparison using TMM-FPKM, followed by log2(x + 1) transformation, as implemented in the edgeR R/Bioconductor package[81].

**GSEA, ssGSEA and score-based gene signatures.** The GSEA and enrichment plots comparing between tumors with high expressions of *ATOH1* and *POU4F3* and those without (Supplementary Fig. 7C) were performed using Qlucore Omics

Explore. Gene pattern from the Broad Institute was used for ssGSEA projection. ssGSEA enrichment scores were computed using the GSVA R/Bioconductor package[82] and gene sets obtained from MSigDB[83]. Additionally, specific gene lists were used to derived ssGSEA scores, including previously described gene sets for the neuroendocrine differentiation[15–17], EMT[49], CCS[84], CAFs[50], APM[63], IIS[60,61], and IFN-γ and expanded immune gene[62] signatures.

We previously developed the repstress gene signature that characterizes the cellular response to replication stress at a functional network level[53]. Briefly, we leveraged cellular characteristics that portend high replication stress such as (i) MYC amplification (ii) sensitivity to CHK1/WEE1 inhibitors (iii) high expression of phosphorylated Chk1, and (iv) high neuroendocrine differentiation in 67 SCLC cell lines, and developed the 17 gene signature (*SRSF1*, *SUV39H1*, *GINS1*, *PRPS1*, *AURKB*, *TNPO2*, *ORC6*, *CCNA2*, *LIG3*, *MTF2*, *GADD45G*, *POLA1*, *POLD4*, *POLE4*, *RFC5*, *RMI1*, *RRM1*). The score was computed as weighted sum of standardized (Z-score) transcript expression for the 17 genes.

**Analysis of differentially regulated biological pathways between NE vs. non-NE SCNCs.** To gain insights into biological pathways underlying the NE phenotypes, we examined differentially regulated biological pathways between NE and non-NE SCNCs. First, Hallmark, Reactome, PID, BioCarta, and KEGG gene sets were retrieved from MSigDB[83] and pathway scores were obtained by ssGSEA for each sample. Applying NE or on-NE differentiation status of each tumor as described above, we identified most differentially regulated 1000 pathways between NE vs. on-NE SCNCs. Initially each pathway scores were z score-normalized among all tumors and stepwisely restricted adjusted P value (obtained by unpaired Student t test followed by Benjamini–Hochberg test), until we identified the most differentially regulated 1000 pathways between NE and non-NE SCNCs using Qlucore Omics Explorer. We then clustered tumors based on the differentially regulated pathway scores by Euclidean distance, complete-linkage non-hierarchical clustering method (Fig. 4d). In addition to previously known pathways upregulated in NE (cell cycle & proliferation, DDR & replication) or non-NE SCNCs (immune response, cell adhesion, metabolism) as indicated in Fig. 4e, we identified 100 co-clustered pathways regulating drug metabolism highly upregulated in specific tumors (Fig. 4d). The upregulation of similar drug metabolism pathways was recently reported in a transcriptional subtype named "NEv2 subtype"[42]. Therefore, we named these pathways as "NEv2-like pathways". To define tumors upregulating these pathways, the ssGSEA-based scores of the 100 pathways were summed up in each tumor and z score-normalized across all the SCNC tumors. Tumors with z-scored NEv2-like pathway summed score >1 were defined as NEv2-like subtype, representing a total of 11 tumors. Lastly, regarding the enrichment of samples obtained from liver lesions in NEv2-like subtype, we evaluated differentially upregulated pathways in samples from liver lesions compared with others by false discovery rate of <1% of unpaired Student t test followed by Benjamini–Hochberg test (Supplementary Table 9).

**CIBERSORT deconvolution.** CIBERSORT is a tool developed by Newman et al.[25] to quantify specific cell types in bulk cell population, using gene expression data. The analyses were run on the CIBERSORT website at http://cibersort.stanford.edu. We applied NE signature[15], LM6, a leukocyte RNA-Seq signature matrix comprised of six peripheral blood subsets[64], and a gene expression signature matrix consisting in four cellular stages of MYC-driven tumor progression[31]. For each run, 100 permutations were performed and quantile normalization was disabled.

**IHC staining and INSM1 histochemical score determination.** IHC stains for synaptophysin (1:20; 790–4407, Roche), chromogranin (1:50; 760–2519, Roche) and INSM1 (1:1,000; sc-271408, Santa Cruz), were performed at National Institutes of Health (NIH), laboratory of Pathology, according to manufacturer's instruction. IHC-stained slides were scanned using the 40X magnification of NanoZoomer S360 Hamamatsu slide scanner. INSM1 H-score was calculated based on the equation: 1 x (% of weakly stained nuclei) + 2 x (% of moderately stained nuclei) + 3 x (% of strongly stained nuclei).

**Hematoxylin and eosin (H&E) staining and neuroendocrine features.** H&E staining was done using standard protocols[85]. Neuroendocrine features morphologically were assessed by a pathologist who was blinded to the results of the neuroendocrine score and were estimated based on nuclear features as well described albeit in cell lines data sets with "classical" morphology as cells with

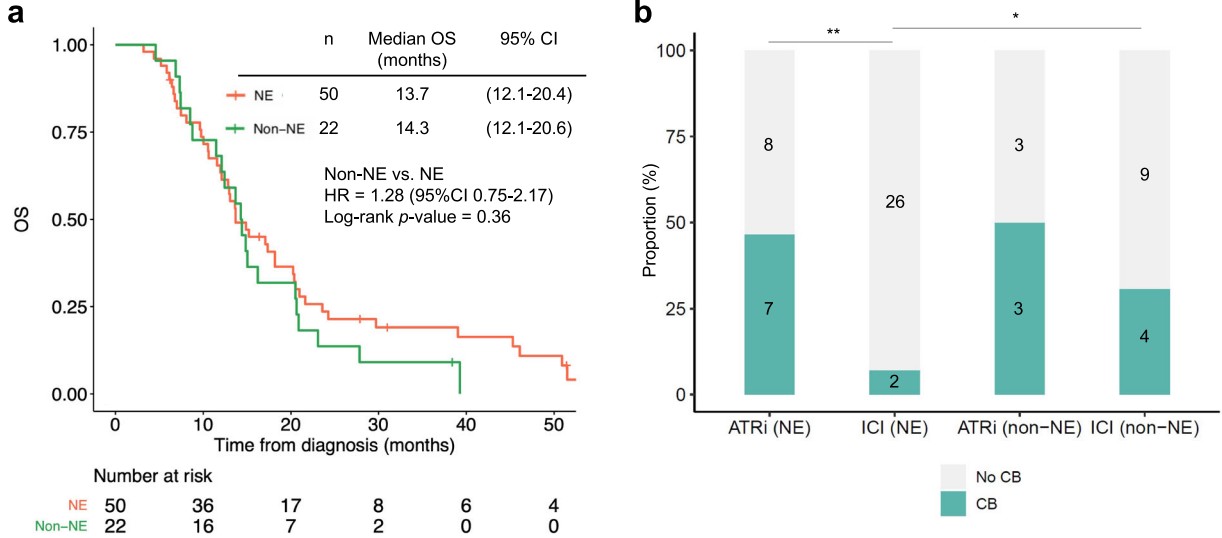

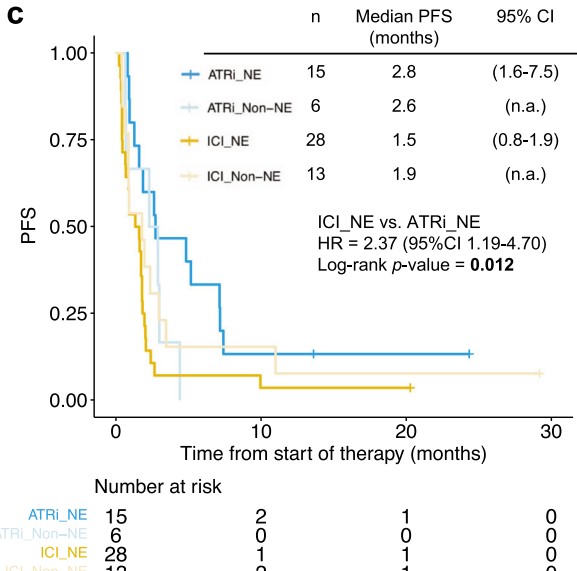

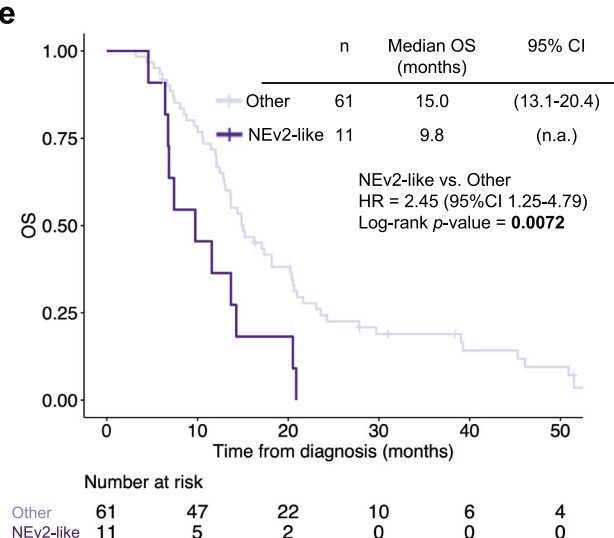

**Cox proportional-hazard analysis of PFS in 41 patients with NE tumors**

| Variable | Univariate | | Multivariate | |
|---|---|---|---|---|
| | HR (95% CI) | P | HR (95% CI) | P |
| Age (Per year) | 0.99 (0.9 - 1.0) | 0.612 | 0.96 (0.9 - 1.0) | 0.163 |
| Sex (Male *vs.* female ) | 0.81 (0.4 - 1.6) | 0.540 | 1.24 (0.6 - 2.6) | 0.559 |
| Smoking (Current/former *vs.* never) | 0.26 (0.0 - 2.0) | 0.195 | 0.38 (0.1 - 3.2) | 0.371 |
| Stage (Extensive *vs.* limited) | 1.67 (0.8 - 3.6) | 0.185 | 1.41 (0.6 - 3.4) | 0.447 |
| Chemotherapy (Resistant *vs.* sensitive) | 0.90 (0.4 - 1.8) | 0.760 | 0.93 (0.4 - 2.0) | 0.855 |
| Systemic treatment (Per line) | 0.74 (0.5 - 1.0) | 0.091 | 0.94 (0.6 - 1.4) | 0.763 |
| Trial (ICI *vs.* ATRi) | 2.83 (1.3 - 4.8) | **0.004** | 2.87 (1.1 – 7.5) | **0.031** |

dense chromatin and absence of a conspicuous nucleoli and "variant" morphology as cells with relatively larger size, clear chromatin, and a conspicuous nucleoli[86].

**CD3+staining and tumor infiltrating lymphocyte (TIL) calculation**. IHC staining for CD3 (pre-diluted; 790–4341, Roche) was done on multiple tissue sections of SCNC cases. IHC-stained slides were scanned using the 40X

magnification of NanoZoomer S360 Hamamatsu slide scanner. TIL were calculated in the intratumoral areas based on previously mentioned consensus guidelines[87] and were objectively assessed using QuPath – an open-source platform for digital pathology imaging analysis[88] and using previously validated settings for melanoma[89]. Quality control of cell segmentation was performed by a pathologist. CD3 + TIL were counted per tumor area millimeter-square and converted into logarithm (base 10) for further correlation studies.

**Fig. 6 Association between SCNC subtypes and response to therapy. a** Kaplan–Meier curves of OS from the date of diagnosis in patients with tumors of NE and non-NE phenotypes. **b** Proportion of patients deriving clinical benefit from ATR inhibition and immunotherapy categorized by NE subtypes. Two-tailed chi-square, **$P = 0.0024$, *$P = 0.0464$. **c** Kaplan–Meier curves of PFS in patients treated with ATR inhibitor and immunotherapy categorized by subtypes. **d** Univariate and multivariate Cox proportional-hazard regression analyses of PFS in patients with NE tumors. **e** Kaplan–Meier curves of OS from the date of diagnosis in patients with NEv2-like subtype and others. HR and p-values by log-rank test are indicated. P-values marked with bold indicate statistical significance. Abbreviations: NE neuroendocrine differentiation; OS overall survival; HR hazard ratio; CI confidence interval; ATRi ataxia telangiectasia and Rad3-related inhibitor; ICI immune checkpoint inhibitor; PFS progression free survival.

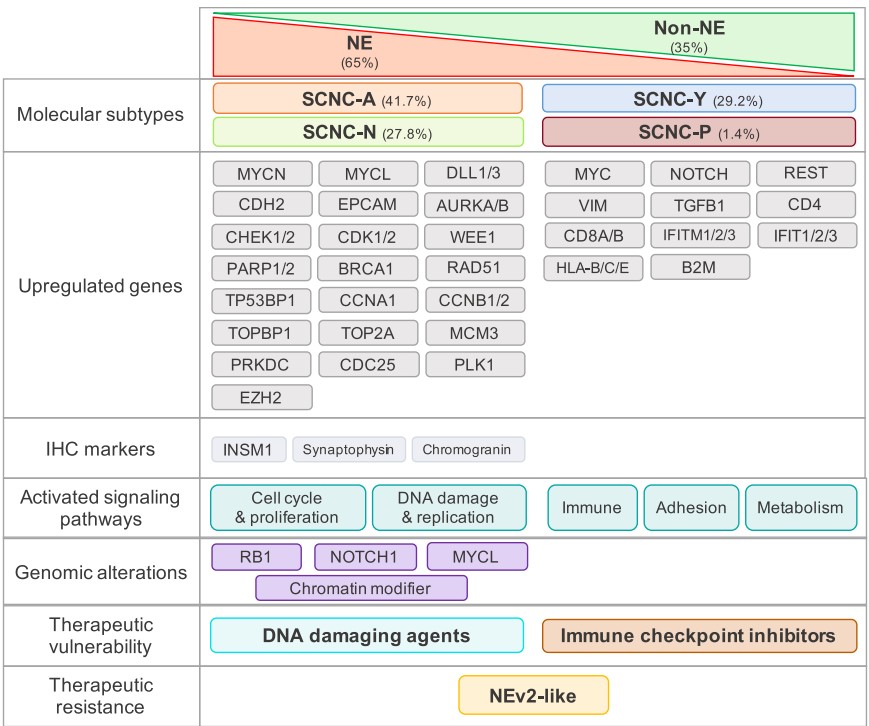

**Fig. 7 Summary of molecular characteristics and potential therapeutic vulnerabilities in NE and non-NE SCNC.** Subtype-specific upregulated genes, IHC markers, signaling pathways, genomic alterations, therapeutic vulnerability, and resistance are displayed.

**Clinical benefit rate at 3 months**. The clinical benefit rate was defined as the percentage of patients who achieved a best overall response of complete response or partial response or stable disease (including unconfirmed partial response) for at least 3 months after start of treatment, according to Response Evaluation Criteria in Solid Tumors (RECIST) v1.1.

**Statistical methods**. R studio version 1.3.1093 (R Foundation for Statistical Computing), GraphPad Prism version 8.1.2 (GraphPad Software), STATA software version 16.0 (Stata-Corp), and Qlucore Omics Explorer version 3.6(2.2) (Qlucore AB) were used to generate figures and statistical analyses. All tests were two-tailed and p-values less than 0.05 were considered significant.

**Reporting summary**. Further information on research design is available in the Nature Research Reporting Summary linked to this article.

## Data availability
The raw data generated in this study (including phenotype, RNA-Seq and WES from human tumors) have been deposited in the database of Genotype and Phenotype (dbGaP) under accession code phs002541.v1.p1 [https://www.ncbi.nlm.nih.gov/projects/gap/cgi-bin/study.cgi?study_id=phs002541.v1.p1&phv=492899&phd=&pha=&pht=11494&phvf=&phdf=&phaf=&phtf=&dssp=1&consent=&temp=1]. The individual-level data are available for download by authorized access only [https://dbgap.ncbi.nlm.nih.gov/aa/wga.cgi?login=&page=login]. The dbGaP request procedure to access individual-level data is described here [https://dbgap.ncbi.nlm.nih.gov/aa/dbgap_request_process.pdf]. Please refer to the release notes for more details [https://ftp.ncbi.nlm.nih.gov/dbgap/studies/phs002541/phs002541.v1.p1/release_notes/Release_Notes.phs002541.SCLC_ChemoRefractory.v1.p1.MULTI.pdf]. There are no limitations to who may access this data, and access will be granted within a

month of request. Source data are provided with this paper. Additional publicly available datasets were used in this study, including early-stage SCLC tumors under accession code EGAS00001000925, PDX (Supplementary Data 4), CDX [https://zenodo.org/record/3574846#.YQLVuI77RPY] and cell line retrieved from CellMiner CDB: Small Cell Lung Cancer [https://discover.nci.nih.gov/SclcCellMinerCDB/]. Source data are provided with this paper.

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

## Acknowledgements

The authors gratefully acknowledge the contributions of the patients who participated in the study. This work was supported by the intramural programs of the Center for Cancer Research, National Cancer Institute (ZIA BC 011793). This work utilized the computational resources of the NIH HPC Biowulf cluster (http://hpc.nih.gov).

## Author contributions

Conceptualization: D.L., N.T., A.T.; Data curation: D.L., N.T., P.D., I.M., C.W.S, V.R., M.J.V, D.M., N.R., S.N., R.V., L.S., Y.C., U.G., A.R., D.A., R.E.M, Z.W.O, A.T.; In vivo experiments: D.A., R.E.M., Z.W.O.; Data analysis: D.L., N.T., V.R.; Figure preparation: D.L., N.T., P.D., I.M.; Manuscript writing: D.L., N.T., A.T. All authors provided input on the manuscript and gave their approval to the final version of the manuscript.

## Funding

## Competing interests

D.L. in an employee of AstraZeneca. A.T. report research funding to the institution from the following entities: EMD Serono, AstraZeneca, Tarveda Therapeutics, Prolynx Inc, and Immunomedics. The other authors have no competing interest to declare. None of the authors have competing non-financial interests.
