## [Peer Review File · Nature Communications]

Heterogeneity of neuroendocrine transcriptional states in metastatic small cell lung cancer tumors and patient-derived modelsREVIEWER COMMENTS

Reviewer #1 (Remarks to the Author): Expert in lung cancer genomics

Lissa, Takahashi, Thomas, and colleagues present a timely and high-quality genomic analysis of small cell lung cancer, providing a large dataset that expands on and provides new insight as to functional differences between molecular subsets of SCLC. Overall, the paper is well-written and the data high-quality and well-interpreted. The main advance in the paper is the description of SCLC subtypes in patient metastatic tumor samples, which is a major advance beyond the SCLC cell lines and resected tumors that the subtypes were originally based on. The authors collected whole exome sequencing data, RNA-seq, and IHC from 72 patients with SCLC or EPSCC. Although not every sample had each of these data collected, RNA-seq appeared to be available for all samples and the other data were presented in such a way that any missing data was not confusing.

A few minor considerations warrant further explanation or attention:

1. p. 7 line 188 – Technically the upstream Hippo pathway suppresses Yap activation. Suggest changing to “activated” to “regulated by the HIPPO...”
2. Figure 2- An obvious explanation for the lower NE score of tumors vs. patient derived models is the low tumor purity of the primary/metastatic samples. This should be discussed as a potential explanation for the trends observed in Fig. 2D-E.
3. p. 11 line 287 – It seems not appropriate to say the NeV2-like tumors lie at the “intersection of the NE and non-NE phenotypes.” The positioning of the dendrogram in Fig 4 is not only the only correct orientation; each clade can be reflected and the data is still valid. I think more important than the positioning of those tumors on the dendrogram is the NeV2-like score indicated in the heatmap. Perhaps cite only that score rather than the position at the “intersection.”
4. Table S4 – It is very difficult to understand the conclusion from this list of enriched pathways. Which pathways are considered important for the NeV2-like tumors? Could a graphical summary of some kind be shown instead (indicating how many pathways were overlapping and which these were). Please also show the average NeV2 score in liver biopsies vs. other sites to support the conclusion that this is not driven by biopsy site.
5. Be consistent about capitalization of Notch vs. NOTCH
6. It would be helpful to expand the discussion of NeV2-like variant and the significance of the present analysis of this possible subtype. This was the less well-explained aspect of the paper.
7. The clinical importance of different response of NE tumors to ICI and ATRi is hugely important. Note to the editor – two papers by the last author are cited and are currently in revision (refs. 13 and 14 cited within). The impact and significance of the present manuscript should likely be evaluated alongside these other related reports.

Reviewer #2 (Remarks to the Author): Expert in bioinformatics and RNA-seq

In this paper, Lissa et al. investigated the molecular characteristics of SCLC through comprehensive genomics and bioinformatics analyses. The focus of the analysis is on comparing neuroendocrine (NE) and non-NE subtypes. Overall, it is a well-written paper that is easy to follow. Genomic sequencing of small cell lung cancer is still understudied at present and I applaud all the efforts made by the team. I only have few technical comments:

- (1) Lines 114-116. I don't think the “advantage” is true here because all GE scores can be normalized/centered when they are calculated and having positive and negative values is just a scale matter.

(2) Please provide the variance explained by PC1 and PC2 in Fig 1d. Also, are PC3 and PC4 not informative at all?

(3) My main questions are in Figs 2 and 3, which are the most important results in this paper. (a) Does the tumor purity have an impact on the results in Fig 2a and 2c? (b) Why only CIBERSORT analysis (requires GEP) has been used here while results from ssGSEA have been considered more robust. (c) Has differential expression (DE) analysis and gene set enrichment analysis (GSEA) been performed on subtypes SCNC-A/N/P/Y? (d) Please use a different color scheme to indicate the loading in Figure 3h. The yellow arrows are very hard to read. (e) I believe the combinatory effect of MYC and Notch pathways can be better explored.

(4) Similar to APM score, NE score should also have medium association/correlation with total Immune infiltration score (such as IIS/TIS). It would be very important to investigate whether the association with APM score (as shown in Fig 4) is confounded or independent. What we know is that almost all immune signatures are correlated somehow and it is vital to tease apart at some level.

(5) It is interesting to see that TMB is significantly higher in NE group. The potential combined effects of TMB and Immune/NS score, (and whether the TMB-IIS/NS subgroups have distinct survivals) can be better investigated in Fig 5/6.

Reviewer #3 (Remarks to the Author): Expert in immunogenomics

The current understanding of molecular subtypes of small cell lung (SCLC) is limited. In this manuscript, the authors sought to define the molecular characteristics of SCLC by performing transcriptomic profiling, DNA sequencing, and histologic assessment on a large cohort of small cell neuroendocrine cancers. The authors found transcriptional subtypes driven by neuroendocrine differentiation and identified intra-tumor heterogeneity of neuroendocrine differentiation in metastatic SCLCs. Of clinical relevance, the authors identified a putative subtype that is characterized by drug resistance.

1. Overall, this study represents an impressive effort that integrates transcriptomic profiling, histology, and clinical data to define molecular subtypes of SCLC. The study is well-designed and the data analysis is thorough. However, the novelty of this study is less clear to this reviewer, and the main conclusions of the study seem to support the existing notion that SCLC transcriptional subtypes are driven by neuroendocrine differentiation.

2. The authors suggest that there is significant intratumoral heterogeneity as transcriptomic analysis suggests there are tumors that exhibit overlap between NE and non-NE signatures (Fig 1B). This is supported by histological findings of both NE and Non-NE in the same tumor (Fig 2b). However, the analysis using CIBERSORT (Fig 2a) to pseudo quantify NE and non-NE proportions seem somewhat circular since NE and Non-NE are defined by the same gene signature, no? A better way to validate the proportion of NE and non-NE each sample would be unbiased, single-cell analyses such as scRNAseq or cytometry. This reviewer recognized that this is likely not feasible since the samples have already been collected, attempts to quantify NE and non-NE in each tumor by histology or other methods should be attempted.

3. The authors suggest that upregulation of Notch target genes and mesenchymal markers (VIM and TGFB) in Non-NE tumors represent a potential epithelial-mesenchymal transition (EMT) process. Since this is a bulk RNAseq data analysis, how can the authors distinguish between EMT vs differences in the composition of the tumor microenvironment (i.e. cancer-associated fibroblasts) between NE and non-NE tumors? Could the difference in mesenchymal genes be caused by a difference in abundance in CAFs, or types of CAFs (iCAF vs myCAF) in these tumors?

4. Similarly, the characterization of the tumor microenvironment relies primarily on bulk tumor RNAsequenign data which lacks cellular resolution. For example, the authors suggest that the "predicted portion of monocytes and natural killer cells were also significantly higher in non-NE relative to NE tumors" This should be validated by some quantification of immune cell populations by immunohistochemistry.

Reviewer #4 (Remarks to the Author): Expert in lung cancer immunotherapy and PDX models

In their manuscript, Lissa and colleagues integrate molecular data from a large set of small cell tumors, mostly from the lung, in an effort to expand upon recent work highlighting the inter-tumoral heterogeneity of small cell lung cancer and its classification into transcriptional subtypes. Some of the findings confirm earlier analyses are relatively well-established, such as the relationship between transcriptional subtype and the defining transcription factors, MYC signaling, NOTCH signaling, EMT, and NE/non-NE gene expression and intra-tumoral heterogeneity. The inclusion of extrapulmonary SCLC tumors is a useful addition which supports their biological similarity to pulmonary SCLC. The inclusion of comprehensive molecular data from metastatic SCLC, including biopsies from multiple metastatic sites, is a useful resource for the SCLC community. Much of the data is connected with clinical outcomes, although issues with those analyses are noted below.

The manuscript does, however, have a number of limitations that reduce the potential impact of the study. Overall most of the findings confirm earlier observations,

Major issues:

(1) The authors elect to follow the SCLC-Y nomenclature and ascribe biological significance to YAP1 expression. However, recent data, such as Baine et al. JTO 2020 and Gay et al. Cancer Cell 2021 provide strong support that YAP1 does not define a unique subgroup of ASCL1-, NeuroD1-, and POU2F3-negative SCLC tumors. Baine et al. did not find YAP1 expression via IHC as a defining feature in the subset of tumors that lacked ASCL1, NEUROD1, and POU2F3. In Figure 2G, the group of tumors that are assigned SCLC-Y often have little to no YAP1 expression (and the few that have robust YAP1 expression appear to be extra-pulmonary). Do the authors consider YAP1 to be the primary transcription factor behind the development of this phenotype/subtype? Did they explore other transcription factors that might define some or all of these tumors?

a. Nomenclature aside, the proportion of tumors that are assigned SCLC-Y is also higher than in other datasets. Do the authors feel that this is due to the fact that they are derived from relapsed disease? How else might they explain this?

(2) Another issue that limits some of the conclusions is the lack of representative POU2F3 samples. The authors find only one such example (~1%), which is well below the proportion seen in other datasets (e.g. Rudin et al., Nat Rev Cancer 2019; Baine et al., JTO 2020; Gay et al., Cancer Cell 2021). As SCLC-P appears to represent a non-NE subtype of SCLC as well, this underrepresentation affects claims made regarding non-NE vs NE SCLC. Of course, the authors could not control this variable, but do they feel that the dearth of SCLC-P samples is indicative of a difference between newly diagnosed and relapsed lung cancers? If so or if not, how might the authors explain the discrepancy compared to other datasets.

(3) The analysis of PDX vs tumor is interesting but since PDX stroma increasingly is derived from the mouse over time, and lacks normal immune infiltration, it is unsurprising that an analysis of human neuroendocrine genes (associated with tumor cells, not stroma) will show a higher neuroendocrine score for PDX models than human specimens. In the CIBERSORT analyses, such as those in related to NE vs non-NE contribution, are the authors utilizing a step to filter out non-malignant cells (e.g. stromal cells). For example, in the comparisons of tumors and PDXs in Figure 2C and D, is the lack of human immune/stromal cells in the PDXs falsely decreasing the non-NE contribution in the xenografts relative to the tumors?

(4) While the authors' use of metastatic and refractory tumors contributes to the novelty of the project, there are several analyses that may benefit from comparison/contrast to existing early stage transcriptional data sets (such as those in George et al. Nature 2015). For example, the various CIBERSORT analyses of NE vs non-NE and the MYC driven signature. Additionally, the same could be done for the SLFN11 and EZH2 comparisons between NE and non-NE, as EZH2 expression is expected to increase and SLFN11 expression decrease following platinum treatment.

(5) Much of the data in Figure 4 and some of the immune checkpoint response data in Figure 6 was recently shown by Owonikoko et al. JTO 2021 and Gay et al. Cancer Cell 2021 (albeit in treatment-naïve metastatic tumors), particularly the connections between non-NE subtypes and immune genes, inflammatory signatures, antigen presenting machinery, and the improved response to immune checkpoint blockade. While it is reassuring to see this data validated further in the relapsed setting here, this leaves a substantial portion of the manuscript without especially

novel findings. The authors make some interesting observations about metabolism, which have not been well investigated, however, and may be better featured in this section.

(6) There is much controversy about RB1 intact SCLC. The authors show that non-NE tumors are less likely to have loss of function mutations in RB1 (although several do have independent copy number gains?). Were there any features of the RB1 intact tumors, such as mixed histology? For example, large-cell NECs have lower rates of RB1 mutations than SCLC, but often have KRAS, or STK11, or KEAP1 mutations instead paired with TP53. Did the authors investigate the presence of any of these mutations here? Were there differences in RB1 RNA expression between the NE and non-NE groups?

(7) The conclusions about benefit from immunotherapy in the non-NE subgroup (also noted in the abstract) do not seem to be adequately supported by the underpowered analysis here. On line 328: The authors report "In contrast, patients with non-NE tumors were more likely to derive clinical benefit from immunotherapy than ATR inhibition (23.1% vs.14.2%)." From Figure 6, it appears that patients treated with ICI in the NE and nonNE had median PFS of 1.7 vs 1.9 months, respectively in Figure 6 which seems unlikely to be a statistically significant difference. The authors should 1. Address whether there is a statistically significant difference in outcomes between the NE and non-NE groups of ICI treated patients, and assuming there is not a significant difference between the groups they should state so explicitly. In addition, they should define what the ICI regimen is, as chemo+immunotherapy should not be grouped with PD1/PDL1 monotherapy or dual PD1/CTLA4 blockade.

(8) Perhaps the most interesting data from the study is the association between response to ATR inhibitors and NE phenotype. For this data, however, the authors should clarify how many of these patients were already reported in the recent Thomas et al paper in Cancer Cell (PMID 3384878) which also reports the association between ATR inhibitor response and the NE phenotype.

Minor issues, typographical errors, etc.:

(1) In Figure 3B, there seems to be a significance marker missing above the MYCL comparison.

(2) In the heatmap in figure 4D, the separated rows at the top would benefit from labels and these may be missing. It is unclear, even with the legend above, what each row here represents.

(3) In Line 301, there is an extra period after "SCLC" that does not belong.

(4) In Line 305, I believe there are typographical errors in the numbers: 10/22 should read 20/22, I believe, and then there is an extra "3" that appears after 6/12 that does not seem to belong. This is based purely on the percentages in the line.

Reviewer #5 (Remarks to the Author): Expert in SCLC therapy and PDX models

This is a well-written and conducted study regarding molecular sequencing analyses of SCLC patient tumors and PDXs with clinical correlation.

My review focused on technical aspects of PDX derivation and Figure 6 at the request of the managing editor.

*PDX technical aspects – described nicely in the methods section.

One data item I recommend including for PDXs that were derived primarily at this institution would be STR data to be included in Table S3 from these 6 NCI-PDX models.

*Figure 6 text:

Pg 12 line 327-328 : "In contrast, patients with non-NE tumors were more likely to derive clinical benefit from immunotherapy than ATR inhibition (23.1% vs. 14.2%)."

Any statistical test performed here? Or, due to limited sample size, underpowered?

If so, should revise to state “may derive greater clinical benefit, however, a statistically significant difference was not observed possibly due underpowered sample size.”

*Figure 6 – PFS analysis seemed to focus quickly on the NE subgroup. It would be helpful to also have PFS and Cox models on all trial comers (both NE and non-NE)

Fig 6B: supplemental of ATR and ICI PFS curves of all-comers would be helpful

Fig 6C: NE v non-NE should be a covariate analyzed in this Cox proportional hazard analysis of PFS or in an additional analyses of all trial patients before the NE-only subgroup shown in 6C – perhaps in supplemental.

REVIEWER COMMENTS

Reviewer #1 (Remarks to the Author): Expert in lung cancer genomics

Lissa, Takahashi, Thomas, and colleagues present a timely and high-quality genomic analysis of small cell lung cancer, providing a large dataset that expands on and provides new insight as to functional differences between molecular subsets of SCLC. Overall, the paper is well-written and the data high-quality and well-interpreted. The main advance in the paper is the description of SCLC subtypes in patient metastatic tumor samples, which is a major advance beyond the SCLC cell lines and resected tumors that the subtypes were originally based on. The authors collected whole exome sequencing data, RNA-seq, and IHC from 72 patients with SCLC or EPSCC. Although not every sample had each of these data collected, RNA-seq appeared to be available for all samples and the other data were presented in such a way that any missing data was not confusing.

A few minor considerations warrant further explanation or attention.

Our response: We thank the reviewer for the favorable feedback on our work. Please see below detailed responses to the specific points.

Point 1 raised by reviewer 1: p. 7 line 188 – Technically the upstream Hippo pathway suppresses Yap activation. Suggest changing to “activated” to “regulated by the HIPPO...”

Our response: We included the reviewer suggestion in the revised manuscript, by changing ‘activated’ to ‘regulated’ (page 7, line 192).

Point 2 raised by reviewer 1: Figure 2- An obvious explanation for the lower NE score of tumors vs. patient derived models is the low tumor purity of the primary/metastatic samples. This should be discussed as a potential explanation for the trends observed in Fig. 2D-E.

Our response: We agree that it is important to clarify the impact of tumor purity on the neuroendocrine score. To address the comment (also raised by reviewers 2 and 4), we estimated tumor purity from tumor samples using bulk RNA-seq and WES data. The abundance of stromal (e.g., cancer-associated fibroblasts, CAFs) and immune cell populations were derived using published gene sets (Bindea et al 2013, Elyada et al 2019) and ssGSEA. CAFs and immune cell type-specific signature scores were negatively correlated with neuroendocrine score. Besides gene expression, we also estimated tumor purity based on DNA copy number alterations on a subset of tumors in our cohort and in independent dataset of early-stage tumors (George et al 2015). Consistent with RNA-seq findings, tumor purity was positively correlated with neuroendocrine score in both cohorts. Supporting these observations, a study of combined SCLC/NSCLC tumors found higher tumor content and lower stromal content in the high NE SCLC component of the tumor compared with the low NE NSCLC component (Quintanal-Villalonga et al 2021). Together, these findings point to transcriptional heterogeneity possibly being shaped by the tumor microenvironment, with higher stromal content as a biological feature of low NE tumors. Thus, while the differences in NE transcriptional programs between patient tumors and patient derived models (Fig. 2d, e) are unlikely to be solely due to the differences in stromal content alone, the contribution of lack of stromal content to the “NE drift” in PDX models cannot be excluded based on our data. Of note, to minimize the effect of tumor-associated stroma replacement by murine stroma, we compared PDXs at passage P1 with their corresponding tumors.

We revised the results and discussion as below. Correlation plots of tumor purity or CAF signature scores vs. neuroendocrine score were added as new Fig S7a, b and new Fig S11c, respectively. The immune infiltration score was included in new Fig S12a.

(page 7, lines 173-176) ‘Of note, tumor purity was positively correlated with the neuroendocrine score suggesting that the paucity of tumor microenvironment (TME) may partly account for the lower neuroendocrine score observed in PDX tumors compared with patient tumors (Fig. S7a, b)’.

(page 16, lines 448-449) ‘The positive correlation between tumor purity and neuroendocrine score would support this last hypothesis.’

Point 3 raised by reviewer 1: p. 11 line 287 – It seems not appropriate to say the NeV2-like tumors lie at the “intersection of the NE and non-NE phenotypes.” The positioning of the dendrogram in Fig 4 is not only the only correct orientation; each clade can be reflected and the data is still valid. I think more important than the positioning of those tumors on the dendrogram is the NeV2-like score indicated in the heatmap. Perhaps cite only that score rather than the position at the “intersection.”

Our response: As suggested by the reviewer, ‘the intersection of the NE and non-NE phenotypes’ was changed to ‘with hybrid NE and non-NE phenotypes’ in the abstract (page 2, lines 33-34), the result section (page 12, line 323-324) and the discussion (page 15, lines 412-413).

Point 4 raised by reviewer 1: Table S4 – It is very difficult to understand the conclusion from this list of enriched pathways.

- a) Which pathways are considered important for the NEv2-like tumors?
- b) Could a graphical summary of some kind be shown instead (indicating how many pathways were overlapping and which these were).
- c) Please also show the average NEv2 score in liver biopsies vs. other sites to support the conclusion that this is not driven by biopsy site.

Our response: We thank the reviewer for pointing out this issue.

a) A list of 100 upregulated pathways in NEv2-like tumors is provided in the revised Table S4 (Table S7 in the revised manuscript). The description in Table S7 was also amended to clarify the list of pathways.

b) None of the 100 enriched pathways in NEv2-like tumors overlap with the 90 pathways specifically upregulated in liver metastases, supporting the conclusion that enrichment of NEv2 tumors in liver biopsy samples is not driven by the biopsy site. We clarified this point in the revised manuscript:

(page 12, lines 329-331) *'Among the 90 pathways specifically upregulated in liver-derived tumors, none were overlapping with NEv2-like pathways (Table S7).'*

c) We evaluated NEv2 score between liver metastases and other biopsy sites. Consistent with the enrichment of liver biopsy samples in NEv2-like subtypes (9/11, 81.8%), the NEv2-like score was significantly higher in liver metastases compared with other biopsy sites ($P < 0.001$ by Mann-Whitney U test). The figure below was added to the revised manuscript (New Fig S12f).

The following text was added to the result section:

(page 12, lines 326-328): *'NEv2-like tumors were more likely to be derived from liver metastases (9/11, 81.8%), and NEv2-like score was significantly higher in liver metastases compared with other biopsy sites (Fig. S12f).'*

Point 5 raised by reviewer 1: Be consistent about capitalization of Notch vs. *NOTCH*

Our response: We revised the nomenclature throughout the manuscript: NOTCH refers to the gene and protein, whilst Notch designates the signaling pathway.

Point 6 raised by reviewer 1: It would be helpful to expand the discussion of NEv2-like variant and the significance of the present analysis of this possible subtype. This was the less well-explained aspect of the paper.

Our response: Thank you. We agree, this is a novel aspect of the study which was not explained well in the initial submission. We have expanded the discussion about the NEv2-like subtype as follows:

(page 16, lines 459-465) *'Finally, we identified a subset of SCNCs characterized by enrichment of xenobiotic and drug transporter pathways, and exceptionally poor response to treatment and survival. Interestingly, the Nev2-like subtype was more frequently identified in liver biopsies. Previous studies have shown that liver metastasis is an independent predictive factor of poor survival in SCLC (Cai et al 2018, Nakazawa et al 2012, Ren et al 2016), in line with the high aggressiveness of the NEv2-like subtype. Further studies are warranted to evaluate core biological characteristics and therapeutic targets in this chemo-resistant subtype.'*

Point 7 raised by reviewer 1: The clinical importance of different response of NE tumors to ICI and ATRi is hugely important. Note to the editor – two papers by the last author are cited and are currently in revision (refs. 13 and 14 cited within). The impact and significance of the present manuscript should likely be evaluated alongside these other related reports.

Our response: As noted in the initial submission and accompanying communications to the editor, references #13 and 14 were under review at the time of the current submission and have since been published (Thomas et al. Cancer Cell. 2021 Apr 12;39(4):566-579.e7 and Roper et al. Nat Commun. 2021 Jun 23;12(1):3880). Both studies report on small patient cohorts treated in clinical trials of ATRi and ICI, and found differential effects on tumor response rates based on tumor NE differentiation. The intent of the current study as outlined in the introduction was to broadly study the impact of molecular characteristics on relapsed SCLC phenotypes, an important question given that the largest series of relapsed SCLC tumors consists of only 18 patients reported by Wagner et al. (Wagner et al 2018). We pooled data from the two individual studies and included additional new data from 29 unique samples from 24 patients. Together the current 100-tumor cohort goes beyond previous studies to provide a well-annotated cohort of mostly relapsed SCLCs, and reveal the complex intra-and intertumoral heterogeneity of NE differentiation of relapsed small cell neuroendocrine tumors, the underlying transcriptional programs that govern these subtypes, and define a novel clinical subtype of chemotherapy-resistant NEv2 subtype.

Reviewer #2 (Remarks to the Author): Expert in bioinformatics and RNA-seq

In this paper, Lissa et al. investigated the molecular characteristics of SCLC through comprehensive genomics and bioinformatics analyses. The focus of the analysis is on comparing neuroendocrine (NE) and non-NE subtypes. Overall, it is a well-written paper that is easy to follow. Genomic sequencing of small cell lung cancer is still understudied at present and I applaud all the efforts made by the team. I only have few technical comments:

Our response: We thank the reviewer for the positive evaluation of our work. We responded to each of the 5 points raised as outlined below.

Point 1 raised by reviewer 2: Lines 114-116. I don't think the "advantage" is true here because all GE scores can be normalized/centered when they are calculated and having positive and negative values is just a scale matter.

Our response: We amended the revised manuscript. The former sentence now reads as follows:

(page 5, lines 116-118) *'The score ranges from -1 to 1, with positive and negative scores respectively indicating NE and non-NE differentiation – a lower negative score providing more confidence that neuroendocrine differentiation is lacking.'*

Point 2 raised by reviewer 2: Please provide the variance explained by PC1 and PC2 in Fig 1d. Also, are PC3 and PC4 not informative at all?

Our response: We used the web resource at <https://nikobshinyapps.shinyapps.io/PCAprjection/> to predict the small cell neuroendocrine feature of our tumor samples, by projecting the RNA-seq data to the PCA framework developed by Balanis et al (Balanis et al 2019). Unfortunately, the percentage of variance explained by PC1 and PC2, and whether PC3 and PC4 were informative is not provided in the original article.

Point 3 raised by reviewer 2: My main questions are in Figs 2 and 3, which are the most important results in this paper.

(a) Does the tumor purity have an impact on the results in Fig 2a and 2c?

Our response: We agree that it is important to clarify the impact of tumor purity on the neuroendocrine score.

To address the reviewer's comment (also raised by reviewers 1 and 4), we estimated tumor purity from bulk tumor samples using RNA-seq and WES data. The abundance of stromal (e.g., cancer-associated fibroblasts, CAFs) and immune cell populations was derived using published gene sets (Bindea et al 2013, Elyada et al 2019) and ssGSEA. We found that CAFs and immune cell type-specific signature scores negatively correlated with neuroendocrine score. Besides gene expression, we also estimated tumor purity based on DNA copy number alterations on a subset of tumors in our cohort and in independent dataset of early-stage tumors (George et al 2015). Consistent with RNA-seq findings, tumor purity was positively correlated with neuroendocrine score in both cohorts. Overall, these results indicate that the differences in tumor purity/variability in TME may in part account for the marked differences in transcriptional programs between patient tumors and model systems, as shown in Fig. 2a and 2c.

We revised the results and discussion as below. Correlation plots of tumor purity or CAF signature scores vs. neuroendocrine score were added as new Fig S7a, b and new Fig S11c, respectively. The immune infiltration score was included in new Fig S12a.

(page 7, lines 173-176) *'Of note, tumor purity was positively correlated with the neuroendocrine score suggesting that the paucity of tumor microenvironment (TME) may partly account for the lower neuroendocrine score observed in PDX tumors compared with patient tumors (Fig. S7a, b).'*

(page 16, lines 448-449) *'The positive correlation between tumor purity and neuroendocrine score would support this last hypothesis.'*

(b) Why only CIBERSORT analysis (requires GEP) has been used here while results from ssGSEA have been considered more robust.

Our response: We agree with the reviewer that results from ssGSEA method are more robust than CIBERSORT deconvolution. However, since the neuroendocrine score is calculated by subtracting the ssGSEA scores of the 25 non-NE genes signature from the 25 NE genes signature, we used the CIBERSORT analysis as another method to show intratumoral heterogeneity. The heatmap below shows similar results to Fig 2a, using NE and non-NE ssGSEA scores.

(c) has differential expression (DE) analysis and gene set enrichment analysis (GSEA) been performed on subtypes SCNC-A/N/P/Y?

Our response: We thank the reviewer for this valuable comment.

We have assessed differential gene expression between SCNC-A, -N, and -Y subtypes (of note, due to the limited number of SCNC-P samples, this subtype was excluded from the analysis). We performed principal component analysis of the top 2000 differentially expressed genes across the three subtypes and observed a distinct separation of the subtypes. We then created three subtype-specific gene signatures from the top 500 contributors to PC1 and PC2. Unsupervised hierarchical clustering analysis using these gene sets revealed three clusters, consistent with the annotated molecular subtypes. We also observed overlap with previously published gene sets (Borromeo et al 2016, Gay et al 2021, Ireland et al 2020, Wooten et al 2019), further supporting the subtype-defining transcriptional signatures.

Furthermore, examination of SCNC-A, -N, and -Y gene sets by gene ontology, revealed enrichment of genes in developmental processes for SCNC-A, DNA damage response and cell cycle for SCNC-N and inflammation and immune response for SCNC-Y, consistent with previous studies (Cai et al 2018, Groves et al 2021, Owonikoko et al 2021, Wooten et al 2019).

The figures below were added to the revised manuscript (New Fig S10a, b) and the following text was added to the result section. In addition, two new supplementary tables were included: subtype-specific gene lists (new Table S4) and enriched GO terms in SCNC subtypes (new Table S5).

(pages 8-9, lines 218-230) *‘To further characterize the distinct transcriptional features of the subtypes, we evaluated the top 2000 differentially expressed genes across SCNC-A, -N, and -Y tumors. A supervised PCA revealed three distinct clusters associated with each molecular subtype (Fig S10a). We created three subtype-specific gene signatures from the top 500 contributors to the first and second principal components (Table S4). Interestingly, NEUROD1 and ASCL1 were among the top contributing genes to PC2. YAP1 was not predicted to contribute as strongly to PC1. However, another core component of the Hippo signaling pathway, LATS2 kinase which directly phosphorylates YAP1, and TGFBR2, whose downstream signaling was shown to interact with the Hippo signaling pathway (Nishio et al 2016, Pefani et al 2016) were among the top 10 negative contributors to PC1. The unsupervised analysis of the subtype-specific gene lists revealed three main clusters, consistent with the expression of the three transcription factors (Fig S10b). We also observed overlap with previously published gene sets (Borromeo et al 2016, Gay et al 2021, Ireland et al 2020, Wooten et al 2019), further supporting the subtype-defining transcriptional signatures.’*

(page 11, lines 314-317) ‘Consistently, gene ontology (GO) analysis of the subtype-specific transcriptional signatures revealed enrichment of DNA damage response and cell cycle related genes in SCNC-N tumors, and inflammation and immune response related genes in SCNC-Y tumors (Table S5).’

(d) Please use a different color scheme to indicate the loading in Figure 3h. The yellow arrows are very hard to read.

Our response: Thank you. The loading color of the PCA plot was changed to grey.

(e) I believe the combinatory effect of MYC and Notch pathways can be better explored.

Our response: To address this reviewer’s comment, we analyzed the expression of components of the Notch signaling pathway and target genes, which were recently identified as MYC targets (Ireland et al 2020). Consistently, MYC target genes (e.g., *NOTCH2*, *HES1*, *-2*, *SOX9*, *JAG1* and *MFNG*) were preferentially expressed in tumors with higher MYC expression. In contrast, the expression of MYC-repressed targets (e.g., *HES6*, *DLL1*, *-4*, *JAG2*, *LFNG*, *LNK1*, *MAML3*, *DLK1* and *FBXW7*) was negatively correlated with MYC expression. These results corroborate the recent findings that MYC and Notch signaling pathway work in concert to promote neuroendocrine cells dedifferentiation (Ireland et al 2020).

The figure below was added as new Fig S11a and the following sentences were included to the revised manuscript:

(page 10, lines 259-260) ‘Several Notch pathway genes identified as MYC targets were differentially regulated, consistent with MYC expression in tumors (Ireland et al 2020) (**Fig S11a**).’

Point 4 raised by reviewer 2: Similar to APM score, NE score should also have medium association/correlation with total Immune infiltration score (such as IIS/TIS). It would be very important to investigate whether the association with APM score (as shown in Fig 4) is confounded or independent. What we know is that almost all immune signatures are correlated somehow and it is vital to tease apart at some level.

Our response: To address the reviewer’s comment, we derived immune cell type-specific scores (Senbabaoglu et al 2016) and computed an immune infiltration score (IIS), as previously established (Bindea et al 2013). The immune signatures were preferentially enriched in non-NE tumors and negatively correlated with neuroendocrine score. A significant association was also observed between the IIS and the APM scores, as previously reported in other cancer types (Wang et al 2019). Of note, genes in APM and IIS signatures did not overlap (Wang et al 2019). These results are in agreement with several papers published recently including (Cai et al 2021, Dora et al 2020, Gay et al 2021, Owonikoko et al 2021, Roper et al 2021).

The heatmap below was included as new Fig S12a:

Point 5 raised by reviewer 2: It is interesting to see that TMB is significantly higher in NE group. The potential combined effects of TMB and Immune/NS score, (and whether the TMB-IIS/NS subgroups have distinct survivals) can be better investigated in Fig 5/6.

Our response: We thank the reviewer for the suggestion. Unfortunately, the small sample size (only 34 patients with TMB data, including 16 treated with ICI) prevented us from properly investigating the survival of subgroups stratified by TMB, immune infiltration score and NE differentiation. Of note, TMB remains controversial at predicting immune checkpoint inhibitor efficacy in SCLC. In a subgroup analysis of IMpower133, a phase III study evaluating the efficacy and safety of adding atezolizumab to standard first line platinum-based chemotherapy, TMB failed to predict benefit in terms of overall survival in TMB-high patients (Horn et al 2018).

To best address the reviewer's comments, we evaluated the TMB and the scores of the immune infiltration (Bindea et al 2013) and the expanded immune (Ayers et al 2017) signatures in patients who benefited or not from ICI. Patients who derived the greatest clinical benefit tended to exhibit a higher level of TMB and a higher score of the two immune signatures. These data were not included in the revised manuscript due to the small sample size undermining the reliability of the analysis.

Reviewer #3 (Remarks to the Author): Expert in immunogenomics

The current understanding of molecular subtypes of small cell lung (SCLC) is limited. In this manuscript, the authors sought to define the molecular characteristics of SCLC by performing transcriptomic profiling, DNA sequencing, and histologic assessment on a large cohort of small cell neuroendocrine cancers. The authors found transcriptional subtypes driven by neuroendocrine differentiation and identified intra-tumor heterogeneity of neuroendocrine differentiation in metastatic SCLCs. Of clinical relevance, the authors identified a putative subtype that is characterized by drug resistance.

Our response: We thank the reviewer for the constructive remarks, for which we provided responses as outlined below.

Point 1 raised by reviewer 3: Overall, this study represents an impressive effort that integrates transcriptomic profiling, histology, and clinical data to define molecular subtypes of SCLC. The study is well-designed and the data analysis is thorough. However, the novelty of this study is less clear to this reviewer, and the main conclusions of the study seem to support the existing notion that SCLC transcriptional subtypes are driven by neuroendocrine differentiation.

Our response: We agree with the reviewer that the notion of SCLC transcriptional subtypes being driven by neuroendocrine differentiation is not new. In fact, this was first described in the 1980s in cell line models. More recently, Rudin et al Nat Rev Cancer 2019 provided a molecular explanation for this heterogeneity, classifying SCLC into high and low neuroendocrine subtypes based on relative expression of lineage-determining transcription factors. Notably, these studies were based on cell lines and resected patient tumors, which may not be representative of most of SCLC, which tend to be unresectable and metastatic at diagnosis. The current study addresses this issue by performing genomic and transcriptomic profiling of a cohort of mostly relapsed SCLCs to assess NE intratumoral heterogeneity, the transcriptional regulators of NE differentiation, the biological features of the subtypes defined by NE differentiation, and potential vulnerabilities of these subtypes. Together, we believe that current study provides the most comprehensive analyses yet of transcriptomic and genetic heterogeneity in metastatic SCLC, along with detailed clinical annotation and outcomes, illustrating the genetic and transcriptional complexity in SCLC patient tumors and provide a rational framework for prospectively targeting the inter-tumoral heterogeneity.

Point 2 raised by reviewer 3: The authors suggest that there is significant intratumoral heterogeneity as transcriptomic analysis suggests there are tumors that exhibit overlap between NE and non-NE signatures (Fig 1B). This is supported by histological findings of both NE and Non-NE in the same tumor (Fig 2b). However, the analysis using CIBERSORT (Fig 2a) to pseudo quantify NE and non-NE proportions seem somewhat circular since NE and Non-NE are defined by the same gene signature, no? A better way to validate the proportion of NE and non-NE each sample would be unbiased, single-cell analyses such as scRNAseq or cytometry. This reviewer recognized that this is likely not feasible since the samples have already been collected, attempts to quantify NE and non-NE in each tumor by histology or other methods should be attempted.

Our response: We agree that unbiased methodologies such as single-cell RNA sequencing, cell sorting by flow cytometry, IHCs, or multiplex immunoassay would be more robust at assessing intratumoral heterogeneity. This limitation was added to the discussion in the revised manuscript.

(pages 16-17, lines 467-471) *‘Our study has several limitations. First, due to the limited tissue availability, the intratumoral heterogeneity, the TME composition and the EMT were inferred computationally through ssGSEA and CIBERSORT deconvolution of bulk RNA-seq data. Although our results are generally in agreement with findings from scRNA-seq (Gay et al 2021, Ireland et al 2020) and SCLC cell lines (Tlemsani et al,2020) which lack the TME, they remain to be validated by unbiased approaches.’*

Based on the reviewer's comment, we used two independent genesets to assess the heterogeneous composition of tumor samples (Gay et al 2021, Groves et al 2021). As shown in the figure below (added to the revised manuscript as new FigS5), SCNC tumors harbor various proportion of SCNC subtypes, supporting the complex intra-tumoral heterogeneity of SCNCs. Nevertheless, the enriched subtype is overall consistent with the neuroendocrine phenotype of the tumor.

The following sentences have been added to the result section:

(page 6, lines 151-152) *'Similar results were obtained when SCLC subtype-specific gene signatures (Gay et al 2021, Groves et al 2021) were applied, with varying subtype proportions noted within each tumor (Fig. S5).'*

Point 3 raised by reviewer 3: The authors suggest that upregulation of Notch target genes and mesenchymal markers (VIM and TGFB) in Non-NE tumors represent a potential epithelial-mesenchymal transition (EMT) process. Since this is a bulk RNAseq data analysis, how can the authors distinguish between EMT vs differences in the composition of the tumor microenvironment (i.e. cancer-associated fibroblasts) between NE and non-NE tumors? Could the difference in mesenchymal genes be caused by a difference in abundance in CAFs, or types of CAFs (iCAF vs myCAF) in these tumors?

Our response: Thank you. As pointed out by the reviewer, the relative contribution of the stromal cells to the EMT score cannot be teased apart from the bulk RNAseq data analysis. Of note, a

negative correlation between EMT and NE differentiation was observed in SCLC cell lines (Tlemsani et al 2020), suggesting that EMT in SCLC is driven at least partly by tumor-intrinsic features. As suggested by the reviewer, we evaluated the abundance of two CAF subpopulations (e.g. iCAFs and myCAFs), using previously published gene signatures (Elyada et al 2019) and ssGSEA. We found that both CAF gene signatures were enriched in non-NE tumors, consistent with the role of CAFs in inducing EMT in cancer cells (Ping et al 2021). Overall, these findings suggest that both tumor cells and CAFs contribute to the lower EMT score observed in non-NE tumors.

The figures below were added as new Fig S11c, and the following text was added to the result section and the discussion:

(page 10, lines 269-272) ‘Of note, we observed the enrichment of two cancer-associated fibroblast (CAF) subpopulation-specific signatures (inflammatory, iCAFs and myofibroblastic, myCAFs) in non-NE tumors (Elyada et al 2019), consistent with the role of CAFs in promoting EMT in cancer cells (Ping et al 2021) (Fig S11c).’

(pages 16-17, lines 467-471) ‘Our study has several limitations. First, due to the limited tissue availability, the intratumoral heterogeneity, the TME composition and the EMT were inferred computationally through ssGSEA and CIBERSORT deconvolution of bulk RNA-seq data. Although our results are in agreement with findings from scRNA-seq (Gay et al 2021, Ireland et al 2020) and SCLC cell lines (Tlemsani et al 2020) which lack the TME, they remain to be validated by unbiased approaches.’

Point 4 raised by reviewer 3: Similarly, the characterization of the tumor microenvironment relies primarily on bulk tumor RNAsequenign data which lacks cellular resolution. For example, the authors suggest that the “predicted portion of monocytes and natural killer cells were also significantly higher in non-NE relative to NE tumors” This should be validated by some quantification of immune cell populations by immunohistochemistry.

Our response: Most of the genomic and transcriptomic profiling in this study were performed on small core needle biopsies, which were used up for bulk RNA-seq and WES. Due to this limitation, we evaluated tumor infiltrating lymphocytes on CD3 stained tissue sections for a subset of tumors. Increased intratumoral TILs were observed in non-NE tumors, as shown in Fig S12d. To extend our analysis of the intratumoral immune landscape, we performed a comprehensive immune profiling using the immune cell-type specific signatures developed by Bindea et al (Bindea et al 2013) and applied ssGSEA. As shown in the figure below (added to the revised manuscript as new FigS12a) we observed increased immune infiltration in non-NE tumors, including a greater gene expression of macrophages, natural killer (NK) cells, T cells and dendritic cells. These findings are consistent with the CIBERSORT deconvolution using the LM6 signature matrix (Chen et al 2018) showing higher proportion NK cells and monocytes in non-NE tumors (Fig S12b, c).

We have revised the result section to include the following sentences:

(page 11, lines 304-309) ‘Correspondingly, gene signatures related to immune cell infiltration (Bindea et al 2013, Senbabaoglu et al 2016), IFN- γ signaling (Ayers et al 2017), and antigen presentation (Wang et al 2019) were negatively correlated with neuroendocrine score (Fig. 4c, S12a). A comprehensive analysis of tumor-infiltrating immune cells using ssGSEA (Bindea et al 2013) and CIBERSORT (Chen et al 2018, Newman et al 2015) revealed the enrichment of several immune cell populations in non-NE relative to NE tumors, including monocytes, macrophages, natural killer (NK) cells, T cells and dendritic cells (Fig. S12a-c) consistent with previous studies (Best et al 2020, Cai et al 2021). The higher T-cell infiltration was confirmed by CD3 IHC staining (Fig. S12d).’

Reviewer #4 (Remarks to the Author): Expert in lung cancer immunotherapy and PDX models

In their manuscript, Lissa and colleagues integrate molecular data from a large set of small cell tumors, mostly from the lung, in an effort to expand upon recent work highlighting the inter-tumoral heterogeneity of small cell lung cancer and its classification into transcriptional subtypes. Some of the findings confirm earlier analyses are relatively well-established, such as the relationship between transcriptional subtype and the defining transcription factors, MYC signaling, NOTCH signaling, EMT, and NE/non-NE gene expression and intra-tumoral heterogeneity. The inclusion of extrapulmonary SCLC tumors is a useful addition which supports their biological similarity to pulmonary SCLC. The inclusion of comprehensive molecular data from metastatic SCLC, including biopsies from multiple metastatic sites, is a useful resource for the SCLC community. Much of the data is connected with clinical outcomes, although issues with those analyses are noted below. The manuscript does, however, have a number of limitations that reduce the potential impact of the study. Overall, most of the findings confirm earlier observations.

Our response: We thank the reviewer for the insightful comments. We have responded to each of the major and minor issues as outlined below.

Major issues:

Major point 1 raised by reviewer 4: The authors elect to follow the SCLC-Y nomenclature and ascribe biological significance to YAP1 expression. However, recent data, such as Baine et al. JTO 2020 and Gay et al. Cancer Cell 2021 provide strong support that YAP1 does not define a unique subgroup of ASCL1-, NeuroD1-, and POU2F3-negative SCLC tumors Baine et al. did not find YAP1 expression via IHC as a defining feature in the subset of tumors that lacked ASCL1, NEUROD1, and POU2F3. In Figure 2g, the group of tumors that are assigned SCLC-Y often have little to no YAP1 expression (and the few that have robust YAP1 expression appear to be extra-pulmonary). Do the authors consider YAP1 to be the primary transcription factor behind the development of this phenotype/subtype? Did they explore other transcription factors that might define some or all of these tumors?

Our response: From the SCNC-Y specific gene signature we derived from a PCA of the top 2000 differentially expressed genes across SCNC-A, -N, and -Y subtypes (see point 3 raised by reviewer 2 for more details), we found that YAP1 was not the most contributing gene to the underlying biology of SCNC-Y. However, we observed that LATS2, another core component of the Hippo signaling pathway, and TGFBR2, whose downstream signaling was shown to interact with YAP1-TAZ (Nishio et al 2016, Pefani et al 2016), were strong contributors to the SCNC-Y phenotype. Accordingly, several other components of the Hippo/YAP1 signaling pathway and downstream targets were enriched in SCNC-Y tumors, including Hippo kinase core (MST1 and MOB1), regulator (NF2), downstream effectors (TAZ, TEAD and SMAD3) and targets (CYR61, CTGF, SMAD7 and RUNX). Overall, these analyses suggest that the Hippo/YAP1 pathway plays a critical role in the development of the SCNC-Y subtype but might not be the main driver, consistent with the findings of two recent papers (Baine et al 2020, Gay et al 2021).

The following sentences and figure (new Fig S8a) were added to the revised manuscript:

(page 7, lines 192-194) *'Expression of YAP1, a transcription factor regulated by the Hippo signaling pathway was higher in tumors with non-NE differentiation (McColl et al 2017), similar to other components of the Hippo pathway (Fig. S8a).'*

(pages 8-9, lines 218-230) ‘To further characterize the distinct transcriptional features of the subtypes, we evaluated the top 2000 differentially expressed genes across SCNC-A, -N, and -Y tumors. A supervised PCA revealed three distinct clusters associated with each molecular subtype (**Fig S10a**). We created three subtype-specific gene signatures from the top 500 contributors to the first and second principal components (**Table S4**). Interestingly, *NEUROD1* and *ASCL1* were among the top contributing genes to PC2. *YAP1* was not predicted to contribute as strongly to PC1. However, another core component of the Hippo signaling pathway, *LATS2* kinase which directly phosphorylates *YAP1*, and *TGFBR2*, whose downstream signaling was shown to interact with the Hippo signaling pathway (Nishio et al 2016, Pefani et al 2016) were among the top 10 negative contributors to PC1. The unsupervised analysis of the subtype-specific gene lists revealed three main clusters, consistent with the expression of the three transcription factors (**Fig S10b**). We also observed overlap with previously published gene sets (Borromeo et al 2016, Gay et al 2021, Ireland et al 2020, Wooten et al 2019), further supporting the subtype-defining transcriptional signatures.’

a. Nomenclature aside, the proportion of tumors that are assigned SCLC-Y is also higher than in other datasets. Do the authors feel that this is due to the fact that they are derived from relapsed disease? How else might they explain this?

Our response: We agree with the reviewer that the higher proportion of SCNC-Y subtype may be related to the disease extent and the prior systemic treatment exposure.

Consistent with a recent publication from Ireland et al. (Ireland et al 2020) showing the MYC-driven temporal tumor cell evolution from SCLC-A to SCLC-N to SCLC-Y, we observed a higher number of SCNC-Y tumors in our cohort of patients with metastatic and recurrent disease compared to a dataset of early-stage tumor samples (George et al 2015). Further, several recent studies have shown that post-chemotherapy SCLCs are more non-NE than treatment naïve tumors (Gay et al 2021, Stewart 2020, Wagner et al 2018). Given that our cohort exclusively contains relapsed samples, a higher proportion of tumors belonging to the non-NE subtype SCNC-Y is expected.

We have added the number of systemic treatments for each patient in Table S2 and extended the discussion with the sentence below:

(page 15, lines 425-429) ‘*A higher frequency of non-NE tumors of the SCNC-Y subtype were observed in our cohort of metastatic and relapsed tumors than previously reported in early-stage tumors, treatment naïve (Baine et al 2020, Gay et al 2021, Rudin et al 2019), supporting the notion that therapy selects for and promotes dynamic evolution of SCLCs to a more non-NE phenotype (Gay et al 2021, Ireland et al 2020, Stewart 2020, Wagner et al 2018).*’

Major point 2 raised by reviewer 4: Another issue that limits some of the conclusions is the lack of representative POU2F3 samples. The authors find only one such example (~1%), which is well below the proportion seen in other datasets (e.g. Rudin et al., Nat Rev Cancer 2019; Baine et al., JTO 2020; Gay et al., Cancer Cell 2021). As SCLC-P appears to represent a non-NE subtype of SCLC as well, this underrepresentation affects claims made regarding non-NE vs NE SCLC. Of course, the authors could not control this variable, but do they feel that the dearth of SCLC-P samples is indicative of a difference between newly diagnosed and relapsed lung cancers? If so or if not, how might the authors explain the discrepancy compared to other datasets.

Our response: We thank this reviewer for pointing out this difference. The lower proportion of SCNC-P tumors in our cohort may be due to the aggressiveness of this subtype. As shown by Gay et al. (Gay et al 2021) SCLC-P subtype is associated with worse overall survival compared with the other transcriptomic subtypes. Therefore, patients with SCNC-P tumors may deteriorate before being considered for second line and later treatments. Supporting this notion, the majority of patients in our cohort had received at least one systemic therapy before tissue sampling (median: 2; range: 1–6). Additionally, the duration between diagnosis and biopsy was relatively longer (median: 7.8 months; range: 0–46.8) suggesting an enrichment of patients with relatively longer disease course in our cohort.

We have added the number of systemic treatments for each patient in Table S2 and extended the discussion with the sentence below:

(page 15, lines 429-430) ‘*However, non-NE tumors of SCNC-P subtype were less frequent, likely due to their exceptionally poor prognosis (Gay et al 2021).*’

Major point 3 raised by reviewer 4: The analysis of PDX vs tumor is interesting but since PDX stroma increasingly is derived from the mouse over time, and lacks normal immune infiltration, it is unsurprising that an analysis of human neuroendocrine genes (associated with tumor cells, not stroma) will show a higher neuroendocrine score for PDX models than human specimens. In the CIBERSORT analyses, such as those in related to NE vs non-NE contribution, are the authors utilizing a step to

filter out non-malignant cells (e.g. stromal cells). For example, in the comparisons of tumors and PDXs in Figure 2C and D, is the lack of human immune/stromal cells in the PDXs falsely decreasing the non-NE contribution in the xenografts relative to the tumors?

Our response: We agree that it is important to evaluate the contribution of the TME to the neuroendocrine score. Of note, we did not filter out stromal and immune cells before performing the CIBERSORT deconvolution. Furthermore, to minimize the effect of tumor stroma replacement by murine stroma throughout PDX passaging, we have compared neuroendocrine score between PDXs at passage P1 and their corresponding tumors.

To address the reviewer's question (also raised by reviewers 1 and 2), we have estimated the proportion of noncancerous cells from bulk tumor samples using RNA-seq data. The abundance of stromal (e.g., cancer-associated fibroblasts, CAFs) and immune cell populations were derived using published gene sets (Bindea et al 2013, Elyada et al 2019) and ssGSEA. We found that CAFs and immune cell type-specific signature scores negatively correlated with neuroendocrine score, suggesting the presence of TME cells in clinical samples. Supporting these observations, a study of combined SCLC/NSCLC tumors found higher tumor content and lower stromal content in the high NE SCLC component of the tumor compared with the low NE NSCLC component (Quintanal-Villalonga et al 2021). Together, these findings point to transcriptional heterogeneity possibly being shaped by the tumor microenvironment, with higher stromal content as a biological feature of low NE tumors. Thus, while the differences in NE transcriptional programs between patient tumors and patient derived models (Fig. 2d, e) are unlikely to be solely due to the differences in stromal content alone, the contribution of lack of stromal content to the "NE drift" in PDX models cannot be excluded based on our data.

We have added the suggested content to the revised result section and discussion. Correlation plots of tumor purity or CAF signature score vs. neuroendocrine score were added as Fig S7a, b or Fig S11c, d, respectively. The IIS score was included in Fig S13a.

(page 7, lines 173-176) *'Of note, tumor purity was positively correlated with the neuroendocrine score suggesting that the paucity of tumor microenvironment (TME) may partly account for the lower neuroendocrine score observed in PDX tumors compared with patient tumors (Fig. S7a, b).'*

(page 16, lines 448-449) *'The positive correlation between tumor purity and neuroendocrine score would support this last hypothesis.'*

Major point 4 raised by reviewer 4: While the authors' use of metastatic and refractory tumors contributes to the novelty of the project, there are several analyses that may benefit from comparison/contrast to existing early-stage transcriptional data sets (such as those in George et al. Nature 2015). For example, the various CIBERSORT analyses of NE vs non-NE and the MYC driven signature. Additionally, the same could be done for the SLFN11 and EZH2 comparisons between NE and non-NE, as EZH2 expression is expected to increase and SLFN11 expression decrease following platinum treatment.

Our response: As suggested by this reviewer, we examined the differences between our cohort and a cohort of early-stage, treatment-naïve SCLCs (George et al 2015). The main findings include:

- An increased proportion of non-NE tumors in our cohort. Consistently, we observed a higher proportion of tumors that harbored a non-NE late time point signature and less tumors enriched with a mid-late time point signature;
- More tumors belonging to the SCNC-Y subtype and less tumors assigned to the SCNC-P subtype in our cohort (see answers to points 1 and 2 raised by this reviewer);
- Non-significant difference in the expression of *SLFN11* and *EZH2* between NE and non-NE early-stage tumors. Unfortunately, due to difficulty of batch correction, we could not directly compare gene expression between the two datasets;

The comparison of MYC-driven time point signatures was included in new Fig S10d. We also added the following sentences to the result section, and extended the discussion:

(pages 9-10, lines 253-256) *Interestingly, in comparison to early-stage, treatment-naïve SCLCs (George et al 2015), the metastatic and relapsed tumors in this cohort harbored lower proportion of the mid-late time point signature and a significantly higher proportion of the late time point signature (Fig. S10d).*

(page 15, lines 425-430) ‘A higher frequency of non-NE tumors of the SCNC-Y subtype were observed in our cohort of metastatic and relapsed tumors than previously reported in early-stage tumors, treatment naïve (Baine et al 2020, Gay et al 2021, Rudin et al 2019), supporting the notion that therapy selects for and promotes dynamic evolution of SCLCs to a more non-NE phenotype (Gay et al 2021, Ireland et al 2020, Stewart 2020, Wagner et al 2018). However, non-NE tumors of SCNC-P subtype were less frequent, likely due to their exceptionally poor prognosis (Gay et al 2021).’

Major point 5 raised by reviewer 4: Much of the data in Figure 4 and some of the immune checkpoint response data in Figure 6 was recently shown by Owonikoko et al. JTO 2021 and Gay et al. Cancer Cell 2021 (albeit in treatment-naïve metastatic tumors), particularly the connections between non-NE subtypes and immune genes, inflammatory signatures, antigen presenting machinery, and the improved response to immune checkpoint blockade. While it is reassuring to see this data validated further in the relapsed setting here, this leaves a substantial portion of the manuscript without especially novel findings. The authors make some interesting observations about metabolism, which have not been well investigated, however, and may be better featured in this section.

Our response: As pointed out by the reviewer, Owonikoko et al. JTO 2021 and Gay et al. Cancer Cell 2021, studies published while the current paper was in submission showed the association between low NE differentiation and tumor immune response. There are however key differences between these recently published studies and the current study in terms of the patients studied, data availability, and new findings.

- As noted by the reviewer, in contrast to the Gay et al study which consisted of treatment naïve metastatic tumors, our cohort consists of a high proportion of relapsed SCLC tumors after platinum-based chemotherapy. Notably, the largest series of SCLC tumors at relapse to date consists of only 18 patients reported by Wagner et al. (Wagner et al 2018). This study to our knowledge represents the largest genomic dataset of patients with relapsed SCLC.

- In contrast to the Gay et al which provides transcriptomic data of only 1302 genes (specifically genes related to SCLC subtype stratification), all transcriptomic data from the current study will be available. Data from Owonikoko et al are not publicly available to our knowledge. One of the key features of our cohort is that the tumors have deep clinical annotation, unlike the previous studies which provide summary data.
- The present study rather than focusing on molecular features associated with immune response, examines a number of other transcriptional programs associated with NE differentiation, and the intratumor heterogeneity of NE characteristics. Together the current 100-tumor cohort goes beyond previous studies to provide a well-annotated cohort of mostly relapsed SCLCs, and reveal the complex intra- and intertumoral heterogeneity of NE differentiation of relapsed small cell neuroendocrine tumors, the underlying transcriptional programs that govern these subtypes, and define a novel clinical subtype of chemotherapy-resistant NEv2 subtype.

As suggested by the reviewer, we further investigated the metabolic pathways which were negatively correlated with NE differentiation. We found that the most significantly enriched metabolic pathways in non-NE tumors included lipid metabolism, nicotinate and nicotinamide metabolism. The KEGG arginine and proline metabolic pathway, whose role in MYC-driven SCLC was recently reported (Chalishazar et al 2019), was also significantly enriched in non-NE tumors. However, the KEGG pathways associated with purine biosynthesis and ribosome biogenesis, also shown to be regulated by MYC in SCLC (Huang et al 2021, Huang et al 2018), were similarly enriched in both tumor subtypes.

The sentences and figure (new Fig S12e) below were added to the revised manuscript. A new Table S6 listing the metabolic pathways is also included.

(pages 11-12, lines 317-323) *‘The most significantly enriched metabolic pathways in non-NE tumors included lipid metabolism, nicotinate and nicotinamide metabolism (Table S6). The KEGG arginine and proline metabolic pathway, whose role in MYC-driven SCLC was recently reported (Chalishazar et al 2019), was also significantly enriched in non-NE tumors (Fig S12e). MYC was also shown to regulate purine biosynthesis and ribosome biogenesis in SCLC (Huang et al 2021, Huang et al 2018), however, the associated KEGG pathways were similarly enriched in both tumor subtypes.’*

Major point 6 raised by reviewer 4: There is much controversy about RB1 intact SCLC. The authors show that non-NE tumors are less likely to have loss of function mutations in RB1 (although several do have independent copy number gains?). Were there any features of the RB1 intact tumors, such as mixed histology? For example, large-cell NECs have lower rates of RB1 mutations than SCLC, but often have KRAS, or STK11, or KEAP1 mutations instead paired with TP53. Did the authors investigate the presence of any of these mutations here? Were there differences in RB1 RNA expression between the NE and non-NE groups?

Our response: Among the eight tumors with intact RB1 (e.g., with neither RB1 mutation nor copy number deletion), none harbored a mutation of KRAS, STK11, or KEAP1 (as shown below and in the revised Fig 5c). Three of these tumors had indeed RB1 copy number gains. The histopathological characteristics of SCNCs were confirmed centrally by the Laboratory of Pathology at the National Institutes of Health.

Additionally, we observed no significant difference in the expression of RB1 between non-NE tumors and NE tumors, as shown below.

Major point 7 raised by reviewer 4: The conclusions about benefit from immunotherapy in the non-NE subgroup (also noted in the abstract) do not seem to be adequately supported by the underpowered analysis here. On line 328: The authors report “In contrast, patients with non-NE tumors were more likely to derive clinical benefit from immunotherapy than ATR inhibition (23.1% vs.14.2%).” From Figure 6, it appears that patients treated with ICI in the NE and nonNE had median PFS of 1.7 vs 1.9 months, respectively in Figure 6 which seems unlikely to be a statistically significant difference. The authors should 1. Address whether there is a statistically significant difference in outcomes between the NE and non-NE groups of ICI treated patients, and assuming there is not a significant difference between the groups they should state so explicitly. In addition, they should define what the ICI regimen is, as chemo+immunotherapy should not be grouped with PD1/PDL1 monotherapy or dual PD1/CTLA4 blockade.

Our response: We thank the reviewer for pointing out this issue (also raised by reviewer 5). There was no significant difference between the median PFS of ICI-treated patients with NE tumors vs. non-NE tumors. Of note, we re-analyzed the treatment efficacy within subgroup after excluding 5 ATRi-treated patients and 1 ICI-treated patient, as tissue sampling was performed at the end of study treatment.

Furthermore, none of the patients received chemotherapy plus immunotherapy. ICI-treated patients either received nivolumab as a single agent or durvalumab plus olaparib. Of note, olaparib is unlikely to add any benefit to the combination, as reported previously (Krebs et al 2017, Thomas et al 2019).

The following sentences were added:

(page 13, lines 359-364) *‘Most patients in our cohort (64/72, 88.9%) were treated at relapse with immune checkpoint inhibitor in monotherapy or in combination with a poly (ADP-ribose) polymerase (PARP) inhibitor (Roper et al 2021, Thomas et al 2019), or ataxia telangiectasia and rad3 related (ATR) inhibitors plus topotecan (Thomas et al 2018, Thomas et al 2021). Five patients received both combinations sequentially. Only patients (60/72, 83.3%) with tissue sampling prior to study treatment were included in the subgroup analysis.’*

(page 13, lines 372-375) *‘In contrast, patients with non-NE tumors tended to derive greater clinical benefit from immunotherapy than ATR inhibition (23.1% vs. 0%). However, the difference was not statistically significant.’*

(pages 17-18, lines 467-472) *‘Our study has several limitations. [...] Secondly, the post-hoc subgroup analysis was largely underpowered and will require further validation in a prospective study.’*

Major point 8 raised by reviewer 4: Perhaps the most interesting data from the study is the association between response to ATR inhibitors and NE phenotype. For this data, however, the authors should clarify how many of these patients were already reported in the recent Thomas et al paper in Cancer Cell (Thomas et al 2021) which also reports the association between ATR inhibitor response and the NE phenotype.

Our response: As noted in the initial submission and accompanying communications to the editor, two manuscripts were under review at the time of the current submission and have since been published (Thomas et al. Cancer Cell. 2021 Apr 12;39(4):566-579.e7 and Roper et al. Nat Commun. 2021 Jun 23;12(1):3880). Both studies report on small patient cohorts treated in clinical trials of ATRi and ICI, and found differential effects on tumor response rates based on tumor NE differentiation. The intent of the current study as outlined in the introduction was to broadly study the impact of molecular characteristics on relapsed SCLC phenotypes, an important question given that the largest series of relapsed SCLC tumors consists of only 18 patients reported by Wagner et al. (Wagner et al 2018). We pooled data from the two individual studies and included additional new data from 29 unique samples from 24 patients. Together the current 100-tumor cohort goes beyond previous studies to provide a well-annotated cohort of mostly relapsed SCLCs, and reveal the complex intra-and intertumoral

heterogeneity of NE differentiation of relapsed small cell neuroendocrine tumors, the underlying transcriptional programs that govern these subtypes, and define a novel clinical subtype of chemotherapy-resistant NeV2 subtype.

Minor issues, typographical errors, etc.:

Minor point 1 raised by reviewer 4: In Figure 3B, there seems to be a significance marker missing above the MYCL comparison.

Our response: The non-significance marker was added to the figure.

Minor point 2 raised by reviewer 4: In the heatmap in figure 4D, the separated rows at the top would benefit from labels and these may be missing. It is unclear, even with the legend above, what each row here represents.

Our response: The row name was added to the heatmap top annotations.

Minor point 3 raised by reviewer 4: In Line 301, there is an extra period after “SCLC” that does not belong.

Our response: The period was removed.

Minor point 4 raised by reviewer 4: In Line 305, I believe there are typographical errors in the numbers: 10/22 should read 20/22, I believe, and then there is an extra “3” that appears after 6/12 that does not seem to belong. This is based purely on the percentages in the line.

Our response: The typographical errors were corrected (page 12, line 344)

Reviewer #5 (Remarks to the Author): Expert in SCLC therapy and PDX models

This is a well-written and conducted study regarding molecular sequencing analyses of SCLC patient tumors and PDXs with clinical correlation.

My review focused on technical aspects of PDX derivation and Figure 6 at the request of the managing editor.

Our response: We thank this reviewer for the positive evaluation of our work. We answered all the comments as outlined below

Point 1 raised by reviewer 5: PDX technical aspects – described nicely in the methods section. One data item I recommend including for PDXs that were derived primarily at this institution would be STR data to be included in Table S3 from these 6 NCI-PDX models.

Our response: We did not perform short tandem repeat (STR) analyses as the provenance of the samples were clear (PDXs were generated from patients under care of clinicians in the author list) and early passage PDX tumors were used.

Point 2 raised by reviewer 5: Figure 6 text:

Pg 12 line 327-328 : “In contrast, patients with non-NE tumors were more likely to derive clinical benefit from immunotherapy than ATR inhibition (23.1% vs. 14.2%).”

Any statistical test performed here? Or, due to limited sample size, underpowered?

If so, should revise to state “may derive greater clinical benefit, however, a statistically significant difference was not observed possibly due underpowered sample size.”

Our response: We thank the reviewer for pointing this out (also raised by reviewer 4). Due to the small sample size, the statistical significance was not reached. Of note, we re-analyzed the treatment efficacy within subgroups after excluding 5 ATRi-treated patients and 1 ICI-treated patient, as tissue sampling was performed at the end of study treatment.

The following sentences were revised in agreement with this reviewer’s suggestion:

(page 13, lines 372-375) *‘In contrast, patients with non-NE tumors tended to derive greater clinical benefit from immunotherapy than ATR inhibition (23.1% vs. 0%). However, the difference was not statistically significant.’*

(page 16-17, lines 467-472) *‘Our study has several limitations. [...] Secondly, the post-hoc subgroup analysis was largely underpowered and will require further validation in a prospective study.’*

Point 3 raised by reviewer 5: Figure 6 – PFS analysis seemed to focus quickly on the NE subgroup. It would be helpful to also have PFS and Cox models on all trial comers (both NE and non-NE)
Fig 6B: supplemental of ATR and ICI PFS curves of all-comers would be helpful

Our response: We thank the reviewer for highlighting this point. A Kaplan-Meier analysis and a multivariate Cox proportional hazard model for PFS on all trial comers were added as new Fig S13a, and S13b.

Multivariate Cox proportional-hazard analysis of PFS from start of study treatment in 60 patients		
Variables	HR (95% CI)	P
Age (Per year)	0.96 (0.93 – 0.99)	0.021
Sex (Male vs. female)	1.26 (0.70 - 2.26)	0.441
Smoking (No vs. yes)	1.02 (0.32 - 3.25)	0.974
Stage (Extensive vs. limited)	1.67 (0.81 - 3.44)	0.168
Disease (EPSCC vs. SCLC)	1.70 (0.68 – 4.25)	0.254
Chemotherapy (Sensitive vs. resistant)	1.17 (0.61 – 2.25)	0.630
Systemic treatment (per line)	0.89 (0.65 - 1.21)	0.452
Study (ATRi vs. ICI)	2.32 (1.17 - 4.59)	0.016
Subtype (NE vs. non-NE)	0.81 (0.39 – 1.67)	0.568

The result section was revised to start with the description of all comers:

(page 13, lines 366-370) ‘There were no significant differences in the clinical characteristics (**Table 1**) or survival between patients with NE and non-NE tumors (median OS, 13.7 vs. 14.3 months respectively; HR, 1.28; 95% CI, 0.75-2.17) (**Fig. 6a**). Patients treated with ICI had significantly shorter PFS than patients receiving ATR inhibitor (median PFS, 1.7 vs. 2.8 months; HR, 1.85; 95% CI, 1.06-3.23, log-rank p-value = 0.03) (**Fig S13a, b**).’

Of note, as some patients were excluded from the analysis due to their biopsy timepoint, we revised the Kaplan-Meier analysis of PFS (revised Fig 6c) and OS (revised Fig S13d), and the Cox proportional hazard model for PFS (revised Fig 6d). Therefore, we also amended the following sentences:

(pages 13-14, lines 378-386) ‘Patients with NE tumors had significantly shorter PFS and a trend towards shorter OS when treated with immunotherapy compared with ATR inhibition (median PFS, 1.5 vs. 2.8 months; HR, 2.37; 95% CI, 1.19-4.70; log-rank p-value = 0.012, **Fig. 6c**; median OS, 2.9 vs. 6.2 months; HR, 1.65; 95% CI, 0.85-3.23; log-rank p-value = 0.14, **Fig. S13d**). Multivariate analysis revealed investigational therapy as the only variable significantly associated with PFS (**Fig. 6d**). Patients with NE tumors were more likely to progress on immunotherapy than with ATR inhibition after adjusting for age, sex, smoking, stage at diagnosis, platinum sensitivity and number of systemic therapies (HR, 2.87; 95% CI, 1.1-7.5; P = 0.031).’

Point 4 raised by reviewer 5: Fig 6C: NE v non-NE should be a covariate analyzed in this Cox proportional hazard analysis of PFS or in an additional analysis of all trial patients before the NE-only subgroup shown in 6C – perhaps in supplemental.

Our response: This analysis is a Cox proportional hazard analysis of PFS in NE patients. Therefore, the NE vs. non-NE covariates is not relevant in this model. A Cox proportional hazard model for PFS on all trial comers including study and neuroendocrine subgroup was added in new FigS13b, as suggested in the previous comment from this reviewer.

References:

- Ayers M, Lunceford J, Nebozhyn M, Murphy E, Loboda A, et al. 2017. IFN-gamma-related mRNA profile predicts clinical response to PD-1 blockade. *J Clin Invest* 127: 2930-40
- Baine MK, Hsieh MS, Lai WV, Egger JV, Jungbluth AA, et al. 2020. SCLC Subtypes Defined by ASCL1, NEUROD1, POU2F3, and YAP1: A Comprehensive Immunohistochemical and Histopathologic Characterization. *J Thorac Oncol* 15: 1823-35
- Balanis NG, Sheu KM, Esedebe FN, Patel SJ, Smith BA, et al. 2019. Pan-cancer Convergence to a Small-Cell Neuroendocrine Phenotype that Shares Susceptibilities with Hematological Malignancies. *Cancer Cell* 36: 17-34 e7
- Best SA, Hess JB, Souza-Fonseca-Guimaraes F, Cursons J, Kersbergen A, et al. 2020. Harnessing Natural Killer Immunity in Metastatic SCLC. *J Thorac Oncol* 15: 1507-21
- Bindea G, Mlecnik B, Tosolini M, Kirilovsky A, Waldner M, et al. 2013. Spatiotemporal dynamics of intratumoral immune cells reveal the immune landscape in human cancer. *Immunity* 39: 782-95
- Borromeo MD, Savage TK, Kollipara RK, He M, Augustyn A, et al. 2016. ASCL1 and NEUROD1 Reveal Heterogeneity in Pulmonary Neuroendocrine Tumors and Regulate Distinct Genetic Programs. *Cell Rep* 16: 1259-72
- Cai H, Wang H, Li Z, Lin J, Yu J. 2018. The prognostic analysis of different metastatic patterns in extensive-stage small-cell lung cancer patients: a large population-based study. *Future Oncol* 14: 1397-407
- Cai L, Liu H, Huang F, Fujimoto J, Girard L, et al. 2021. Cell-autonomous immune gene expression is repressed in pulmonary neuroendocrine cells and small cell lung cancer. *Commun Biol* 4: 314
- Chalishazar MD, Wait SJ, Huang F, Ireland AS, Mukhopadhyay A, et al. 2019. MYC-Driven Small-Cell Lung Cancer is Metabolically Distinct and Vulnerable to Arginine Depletion. *Clin Cancer Res* 25: 5107-21
- Chen B, Khodadoust MS, Liu CL, Newman AM, Alizadeh AA. 2018. Profiling Tumor Infiltrating Immune Cells with CIBERSORT. *Methods Mol Biol* 1711: 243-59
- Dora D, Rivard C, Yu H, Bunn P, Suda K, et al. 2020. Neuroendocrine subtypes of small cell lung cancer differ in terms of immune microenvironment and checkpoint molecule distribution. *Mol Oncol* 14: 1947-65
- Elyada E, Bolisetty M, Laise P, Flynn WF, Courtois ET, et al. 2019. Cross-Species Single-Cell Analysis of Pancreatic Ductal Adenocarcinoma Reveals Antigen-Presenting Cancer-Associated Fibroblasts. *Cancer Discov* 9: 1102-23
- Gay CM, Stewart CA, Park EM, Diao L, Groves SM, et al. 2021. Patterns of transcription factor programs and immune pathway activation define four major subtypes of SCLC with distinct therapeutic vulnerabilities. *Cancer Cell* 39: 346-60 e7
- George J, Lim JS, Jang SJ, Cun Y, Ozretic L, et al. 2015. Comprehensive genomic profiles of small cell lung cancer. *Nature* 524: 47-53
- Groves SM, Ireland AS, Liu Q, Simmons AJ, Lau K, et al. 2021. Cancer Hallmarks Define a Continuum of Plastic Cell States between Small Cell Lung Cancer Archetypes. *bioRxiv*
- Horn L, Mansfield AS, Szczesna A, Havel L, Krzakowski M, et al. 2018. First-Line Atezolizumab plus Chemotherapy in Extensive-Stage Small-Cell Lung Cancer. *N Engl J Med* 379: 2220-29
- Huang F, Huffman KE, Wang Z, Wang X, Li K, et al. 2021. Guanosine triphosphate links MYC-dependent metabolic and ribosome programs in small-cell lung cancer. *J Clin Invest* 131

- Huang F, Ni M, Chalise MD, Huffman KE, Kim J, et al. 2018. Inosine Monophosphate Dehydrogenase Dependence in a Subset of Small Cell Lung Cancers. *Cell Metab* 28: 369-82 e5
- Ireland AS, Micinski AM, Kastner DW, Guo B, Wait SJ, et al. 2020. MYC Drives Temporal Evolution of Small Cell Lung Cancer Subtypes by Reprogramming Neuroendocrine Fate. *Cancer Cell*
- Krebs M, Ross K, Kim S, De Jonge M, Barlesi F, et al. 2017. P1.15-004 An Open-Label, Multitumor Phase II Basket Study of Olaparib and Durvalumab (MEDIOLA): Results in Patients with Relapsed SCLC. *J Thorac Oncol.* 12: S2044-S45
- McColl K, Wildey G, Sakre N, Lipka MB, Behtaj M, et al. 2017. Reciprocal expression of INSM1 and YAP1 defines subgroups in small cell lung cancer. *Oncotarget* 8: 73745-56
- Nakazawa K, Kurishima K, Tamura T, Kagohashi K, Ishikawa H, et al. 2012. Specific organ metastases and survival in small cell lung cancer. *Oncol Lett* 4: 617-20
- Newman AM, Liu CL, Green MR, Gentles AJ, Feng WG, et al. 2015. Robust enumeration of cell subsets from tissue expression profiles. *Nat Methods* 12: 453-7
- Nishio M, Sugimachi K, Goto H, Wang J, Morikawa T, et al. 2016. Dysregulated YAP1/TAZ and TGF-beta signaling mediate hepatocarcinogenesis in Mob1a/1b-deficient mice. *Proc Natl Acad Sci U S A* 113: E71-80
- Owonikoko TK, Dwivedi B, Chen Z, Zhang C, Barwick B, et al. 2021. YAP1 Expression in SCLC Defines a Distinct Subtype With T-cell-Inflamed Phenotype. *J Thorac Oncol* 16: 464-76
- Pefani DE, Pankova D, Abraham AG, Grawenda AM, Vlahov N, et al. 2016. TGF-beta Targets the Hippo Pathway Scaffold RASSF1A to Facilitate YAP/SMAD2 Nuclear Translocation. *Mol Cell* 63: 156-66
- Ping Q, Yan R, Cheng X, Wang W, Zhong Y, et al. 2021. Cancer-associated fibroblasts: overview, progress, challenges, and directions. *Cancer Gene Ther*
- Quintanal-Villalonga A, Taniguchi H, Zhan YA, Hasan MM, Chavan SS, et al. 2021. Multi-omic analysis of lung tumors defines pathways activated in neuroendocrine transformation. *Cancer Discov*
- Ren Y, Dai C, Zheng H, Zhou F, She Y, et al. 2016. Prognostic effect of liver metastasis in lung cancer patients with distant metastasis. *Oncotarget* 7: 53245-53
- Roper N, Velez M, Chiappori A, Kim YS, Wei JS, et al. 2021. Notch signaling and efficacy of PD-1/PD-L1 blockade in relapsed small cell lung cancer. *Nat Commun* 12
- Rudin CM, Poirier JT, Byers LA, Dive C, Dowlati A, et al. 2019. Molecular subtypes of small cell lung cancer: a synthesis of human and mouse model data. *Nat Rev Cancer* 19: 289-97
- Senbabaoglu Y, Gejman RS, Winer AG, Liu M, Van Allen EM, et al. 2016. Tumor immune microenvironment characterization in clear cell renal cell carcinoma identifies prognostic and immunotherapeutically relevant messenger RNA signatures. *Genome Biol* 17: 231
- Stewart ACG, C.M.; 2020. Single-cell analyses reveal increased intratumoral heterogeneity after the onset of therapy resistance in small-cell lung cancer. *Nature* 1: 423-36
- Thomas A, Redon CE, Sciuto L, Padiernos E, Ji J, et al. 2018. Phase I Study of ATR Inhibitor M6620 in Combination With Topotecan in Patients With Advanced Solid Tumors. *J Clin Oncol* 36: 1594-602
- Thomas A, Takahashi N, Rajapakse VN, Zhang X, Sun Y, et al. 2021. Therapeutic targeting of ATR yields durable regressions in small cell lung cancers with high replication stress. *Cancer Cell* 39

- Thomas A, Vilimas R, Trindade C, Erwin-Cohen R, Roper N, et al. 2019. Durvalumab in Combination with Olaparib in Patients with Relapsed SCLC: Results from a Phase II Study. *J Thorac Oncol* 14: 1447-57
- Tlemsani C, Pongor L, Elloumi F, Girard L, Huffman KE, et al. 2020. SCLC-CellMiner: A Resource for Small Cell Lung Cancer Cell Line Genomics and Pharmacology Based on Genomic Signatures. *Cell Rep* 33: 108296
- Wagner AH, Devarakonda S, Skidmore ZL, Krysiak K, Ramu A, et al. 2018. Recurrent WNT pathway alterations are frequent in relapsed small cell lung cancer. *Nat Commun* 9: 3787
- Wang S, He Z, Wang X, Li H, Liu XS. 2019. Antigen presentation and tumor immunogenicity in cancer immunotherapy response prediction. *Elife* 8
- Wooten DJ, Groves SM, Tyson DR, Liu Q, Lim JS, et al. 2019. Systems-level network modeling of Small Cell Lung Cancer subtypes identifies master regulators and destabilizers. *PLoS Comput Biol* 15: e1007343

REVIEWER COMMENTS

Reviewer #1 (Remarks to the Author):

The authors have sufficiently addressed my comments.

Reviewer #2 (Remarks to the Author):

All my comments have been addressed.

Reviewer #3 (Remarks to the Author):

The authors have adequately addressed all of my comments and concerns.

Reviewer #4 (Remarks to the Author):

The revised manuscript by Lissa et al and rebuttal address many of the issues raised by my review and others, and have improved the manuscript significantly, but several important issues remain to be fully addressed. These include:

1. The statement in the abstract amplified in the main text “in contrast to non-NE tumors which were more likely to respond to immune response targeted therapies” is unclear and does not appear to be supported by the data. Are non-NE tumors more likely to respond to immune response targeted therapies than to replication stress-targeted therapies? Or are non-NE tumors more likely to respond to immunotherapy than NE tumors? In any case, it does not seem that either comparison shows a statistically significant difference.

2. The comparisons of treatment response remain overstated in several ways.

a. For comparison of clinical benefit, they define “partial response or stable disease >4months”.

This is not a standard definition of clinical benefit and in the clinical report of the topotecan combo (Thomas et al JCO 2018) they reported response rate and SD >3m. For the comparisons of benefit, standard criteria (ORR and PFS) should be the primary comparisons.

b. For the immunotherapy analyses, durva plus olaparib should not be included in the immunotherapy arm. (The authors note that the combination of olaparib plus durva did not show significant benefit, but SCLC subgroup-specific differences in sensitivity to PARP inhibitors including olaparib have been observed.)

Furthermore, for the ATRi arm, the number receiving ATRi plus topotecan or other ATRi combo should be specified (and ideally the combination with topotecan should be considered as a separate group). The discussion of this arm emphasizes only ATRi but of course topotecan is active in this setting therefore not possible to determine the relative impact of the two drugs. For example:

“Patients with NE tumors had significantly shorter PFS and a trend towards shorter OS when treated with immunotherapy compared with ATR inhibition” (page 12-13).

3. Regarding the statement in the abstract and similar ones elsewhere in the manuscript:

“Transcriptomic analysis confirmed previously described subtypes based on ASCL1, NEUROD1, POU2F3, YAP1 and ATOH1 expression”. As noted in the earlier review, multiple recent studies (including Baine et al, JTO 2020 from the group that initially proposed the YAP1 subgroup and Gay et al, Cancer Cell 2021) have established that YAP1 does not define one of the distinct major subgroups of SCLC. Therefore, that statement that this analysis confirms the presence of this subgroup should be modified throughout; while YAP1 is present in SCLC, it clearly is present in several of the subgroups, mainly the non-NE subtypes.

The response of the authors misses this point: “Overall, these analyses suggest that the

Hippo/YAP1 pathway plays a critical role in the development of the SCNC-Y subtype but might not be the main driver, consistent with the findings of two recent papers (Baine et al 2020, Gay et al 2021).” This issue isn’t that YAP1 isn’t the main driver of the YAP1 subgroup, it is that there is not a unique YAP1 subgroup (it is slightly increased in the SCLC-I subgroup in Gay et al, which shows increased EMT). The authors should remove the references to the YAP1 subgroup, although they certainly could highlight the role of the YAP/Hippo pathway itself as the data dictates.

4. The issue of overrepresentation of the NE subtypes potentially being related to the reduced human stroma in PDXs is an important one; the authors do acknowledge it in the revision but it remains relevant.

Also, regarding the statement: “Of note, tumor purity was positively correlated with the neuroendocrine score suggesting that the paucity of tumor microenvironment (TME) may partly account for the lower neuroendocrine score observed in PDX tumors compared with patient tumors (Fig. S7a, b)’. It seems this statement should be “higher neuroendocrine score” instead of lower.

Reviewer #5 (Remarks to the Author):

The authors responded satisfactorily to my comments with appropriate revisions.

REVIEWER #4 COMMENTS

The revised manuscript by Lissa et al and rebuttal address many of the issues raised by my review and others, and have improved the manuscript significantly, but several important issues remain to be fully addressed. These include:

Point 1: The statement in the abstract amplified in the main text “in contrast to non-NE tumors which were more likely to respond to immune response targeted therapies” is unclear and does not appear to be supported by the data. Are non-NE tumors more likely to respond to immune response targeted therapies than to replication stress-targeted therapies? Or are non-NE tumors more likely to respond to immunotherapy than NE tumors? In any case, it does not seem that either comparison shows a statistically significant difference.

Our response: We thank the reviewer for this comment. Please see response to point 2a which addresses a similar point.

Point 2a: The comparisons of treatment response remain overstated in several ways.
a. For comparison of clinical benefit, they define “partial response or stable disease>4months”. This is not a standard definition of clinical benefit and in the clinical report of the topotecan combo (Thomas et al JCO 2018) they reported response rate and SD>3m. For the comparisons of benefit, standard criteria (ORR and PFS) should be the primary comparisons.

Our response: As noted in the Response Evaluation Criteria in Solid Tumors (RECIST) guidelines, there are no standard definitions of clinical benefit, and the clinical relevance of the duration of SD varies for different cancer types and tumor grades (Therasse et al 2000). The JCO 2018 paper was a phase I clinical trial that included treatment refractory patients with a range of tumour types, and thus used a less stringent 3-month threshold to define clinical benefit.

Further, the interval between tumor assessments was different for each trial: radiographic evaluations were repeated every two cycles of 21 days for the combination of berzosertib and topotecan (NCT024870), every two cycles of 28 days for the combination of durvalumab and olaparib (NCT02484404), and every 3 cycles of 14 days for nivolumab as monotherapy.

As suggested by this reviewer, we assessed the clinical benefit rate defined as the percentage of patients who achieved CR or PR as best response or SD (including unconfirmed PR) for at least 3 months after start of treatment. As shown in the barplot below (new Figure 6b), patients with NE tumors derived greater clinical benefit from ATR inhibition than immunotherapy (46.7% vs. 7.1%; Chi-square, two tailed $P = 0.0024$). Furthermore, ICI-treated patients with non-NE tumors achieved higher benefit than patients with NE tumors (30.8% vs. 7.1%; Chi-square, two tailed $P = 0.046$).

We also calculated the ORR and observed a similar trend (new Figure S13d). Thus, a higher ORR was observed in patients with NE tumors treated with ATR inhibitor compared to ICI (33.3% vs. 7.1%; Chi-square, two tailed $P = 0.027$).

The median PFS were already compared in previous versions of the manuscript and reported in Fig 6c.

In addition to the new figures, the following sentences were revised throughout the manuscript:

(page 2, lines 34-38) '*NE tumors [...] were more likely to respond to replication stress-targeted therapies. In contrast, patients preferentially benefited from immunotherapy if their tumors were non-NE.*

(page 13, lines 370-378) '*Among ICI-treated patients, those with non-NE tumors derived the greater clinical benefit compared to NE tumors (30.8% vs. 7.1%; Chi-square, $P = 0.046$)*

(Fig. 6b). [...] Patients with NE tumors had higher overall response and clinical benefit when treated with ATR inhibitor compared to immunotherapy (ORR: 33.3% vs. 7.1%; Chi-square, $P = 0.027$ and clinical benefit rate: 46.7% vs. 7.1%; Chi-square, $P = 0.0024$) (**Fig. 6b, S13c**).'

(page 16, lines 454-457) 'However, exploration of the predictive significance of neuroendocrine subtype classification for investigational therapies revealed that NE tumors preferentially respond to replication-stress targeted therapies, while ICI-treated patients with non-NE tumors derived greater benefit compared to those with NE tumors.'

(page 30, lines 746-750)

'Clinical benefit rate at 3 months

The clinical benefit rate was defined as the percentage of patients who achieved a best overall response of complete response or partial response in the first 3 months of treatment, or stable disease (including unconfirmed partial response) for at least 3 months after start of treatment, according to Response Evaluation Criteria in Solid Tumors (RECIST) v1.1.'

Point 2b: For the immunotherapy analyses, durva plus olaparib should not be included in the immunotherapy arm. (The authors note that the combination of olaparib plus durva did not show significant benefit, but SCLC subgroup-specific differences in sensitivity to PARP inhibitors including olaparib have been observed.)

Furthermore, for the ATRi arm, the number receiving ATRi plus topotecan or other ATRi combo should be specified (and ideally the combination with topotecan should be considered as a separate group). The discussion of this arm emphasizes only ATRi but of course topotecan is active in this setting therefore not possible to determine the relative impact of the two drugs. For example:

"Patients with NE tumors had significantly shorter PFS and a trend towards shorter OS when treated with immunotherapy compared with ATR inhibition" (page 12-13).

Our response: We respectfully disagree with the suggestion to not include efficacy results of PARPi combinations in the immunotherapy cohort. The lack of synergistic interaction of PARP inhibition and immune checkpoint blockade has now been documented in several cancers, including SCLC (Takahashi et al 2020, Thomas et al 2019). The SCLC subgroup-specific differences in sensitivity to PARP inhibitors pointed out by the reviewer were documented in preclinical models but have not been found in patients, where PARPi monotherapy have marginal efficacy at best (de Bono et al 2017). Finally, the two immunotherapy-treated cohorts (nivolumab as monotherapy and durvalumab in combination with olaparib) were combined previously in a subgroup analysis of patients deriving clinical benefit from immunotherapy. The article was recently published in Nature Communications (Roper et al 2021).

The ATRi arm only includes the combination of berzosertib and topotecan (ClinicalTrials.gov # NCT024870). The 's' was removed from 'rad3 related(ATR) inhibitors plus topotecan' (page 13, line 362) to eliminate any confusion.

Point 3: Regarding the statement in the abstract and similar ones elsewhere in the manuscript: "Transcriptomic analysis confirmed previously described subtypes based on ASCL1, NEUROD1, POU2F3, YAP1 and ATOH1 expression". As noted in the earlier review, multiple recent studies (including Baine et al, JTO 2020 from the group that initially proposed the YAP1 subgroup and Gay et al, Cancer Cell 2021) have established that YAP1 does not define one of the distinct major subgroups of SCLC. Therefore, that statement that this analysis confirms the presence of this subgroup should be modified throughout; while YAP1 is present in SCLC, it clearly is present in several of the subgroups, mainly the non-NE subtypes.

The response of the authors misses this point: “Overall, these analyses suggest that the Hippo/YAP1 pathway plays a critical role in the development of the SCNC-Y subtype but might not be the main driver, consistent with the findings of two recent papers (Baine et al 2020, Gay et al 2021).” This issue isn’t that YAP1 isn’t the main driver of the YAP1 subgroup, it is that there is not a unique YAP1 subgroup (it is slightly increased in the SCLC-I subgroup in Gay et al, which shows increased EMT). The authors should remove the references to the YAP1 subgroup, although they certainly could highlight the role of the YAP/Hippo pathway itself as the data dictates.

Our response: We acknowledge that the role of YAP1 as a subtype-defining marker remains controversial. However, we respectfully disagree to remove the references to the YAP1 subgroup throughout the manuscript. Contrary to the two recent articles highlighted by the reviewer, which reported the absence or minimal expression of *YAP1* in SCLC by RNA-seq and IHC, *YAP1* expression defines a distinct subtype in our cohort as shown in Fig 2h, 2g, S8b and S8d. It is mainly expressed in non-NE tumors and rarely co-expressed with the other transcriptional drivers, contrary to *ASCL1* and *NEUROD1* (Fig 2f, 2g and S8b). Our results are concordant with a recent publication reporting the SCLC-Y subtype in a cohort of 59 patients with pulmonary neuroendocrine tumors, including 34 SCLCs (Owonikoko et al 2021). As requested by this reviewer in the previous revision (point 1), we further explored other subtype-defining transcription factors and could not identify a more specific one than the Hippo/YAP1 pathway. As stated by Baine et al, JTO, 2020: ‘Overall, the role of YAP1 as a subtype-defining marker in SCLC will require clarification in future studies.’

Point 4: The issue of overrepresentation of the NE subtypes potentially being related to the reduced human stroma in PDXs is an important one; the authors do acknowledge it in the revision but it remains relevant.

Also, regarding the statement: “Of note, tumor purity was positively correlated with the neuroendocrine score suggesting that the paucity of tumor microenvironment (TME) may partly account for the lower neuroendocrine score observed in PDX tumors compared with patient tumors (Fig. S7a, b)”. It seems this statement should be “higher neuroendocrine score” instead of lower.

Our response: We thank the reviewer for identifying this error. The text was modified as follows:

(page 7, lines 173-176): ‘Of note, tumor purity was positively correlated with the neuroendocrine score suggesting that the paucity of tumor microenvironment (TME) may partly account for the *higher* neuroendocrine score observed in PDX tumors compared with patient tumors (Fig. S7a, b)’

REFERENCES

- de Bono J, Ramanathan RK, Mina L, Chugh R, Glaspy J, et al. 2017. Phase I, Dose-Escalation, Two-Part Trial of the PARP Inhibitor Talazoparib in Patients with Advanced Germline BRCA1/2 Mutations and Selected Sporadic Cancers. *Cancer Discov* 7: 620-29
- Owonikoko TK, Dwivedi B, Chen Z, Zhang C, Barwick B, et al. 2021. YAP1 Expression in SCLC Defines a Distinct Subtype With T-cell-Inflamed Phenotype. *J Thorac Oncol* 16: 464-76
- Roper N, Velez MJ, Chiappori A, Kim YS, Wei JS, et al. 2021. Notch signaling and efficacy of PD-1/PD-L1 blockade in relapsed small cell lung cancer. *Nat Commun* 12: 3880

- Takahashi N, Surolia I, Thomas A. 2020. Targeting DNA Repair to Drive Immune Responses: It's Time to Reconsider the Strategy for Clinical Translation. *Clin Cancer Res* 26: 2452-56
- Therasse P, Arbuck SG, Eisenhauer EA, Wanders J, Kaplan RS, et al. 2000. New guidelines to evaluate the response to treatment in solid tumors. European Organization for Research and Treatment of Cancer, National Cancer Institute of the United States, National Cancer Institute of Canada. *J Natl Cancer Inst* 92: 205-16
- Thomas A, Vilimas R, Trindade C, Erwin-Cohen R, Roper N, et al. 2019. Durvalumab in Combination with Olaparib in Patients with Relapsed SCLC: Results from a Phase II Study. *J Thorac Oncol* 14: 1447-57

REVIEWERS' COMMENTS

Reviewer #4 (Remarks to the Author):

In the second revision of the manuscript by Lissa et al, and their second rebuttal, they have reasonably addressed most of the issues I previously raised, but there remain one moderately important and one very important issue that have not been adequately addressed. The moderately important issue is that, as raised earlier, the combination of PARPi/ICB (or topotecan/ATR inhibitor) should be presented separately from PD1 monotherapy or ATR monotherapy; all of these drugs have monotherapy activity (including PARP inhibitors) and predictors of the combination may or may not predict either monotherapy (especially since PARPi and ICB are thought to have different predictors).

The more important issue not addressed is that YAP1 is not a distinct subgroup separate from established ASCL1, NeuroD1 and, to a lesser extent, Pou2F3 subgroups. There is compelling evidence from multiple independent groups that three markers define predominantly non-overlapping, biologically distinct subgroups (ASCL1, NeuroD1, and POU2F3), although the fourth distinct group has been the subject of some debate. In a review article, (Rudin et al Nat Rev Cancer) it was suggested YAP1 may define a fourth subgroup but in multiple studies investigating this issue since that time from Rudin et al and others (Baine et al, JTO, 2020; Gay et al, Cancer Cell 2021; Qu et al, JTO, 2021) it is clear and unequivocal that, while YAP1 is expressed in some SCLC cases, it does not define a distinct subgroup but rather overlaps with the others.

To illustrate this in more detail: from the recent paper on which Dr. Thomas is an author (Qu et al, JTO, 2021), the frequency of YAP1 staining is below (Supplemental figure S4). Here, YAP1 is expressed without the other markers in only 2.1% of the cases (3/142), but is expressed along with ASCL1, NEUROD1, or POU2F3 in 10 cases (7%). In other words, of the 13 cases of YAP1 expression, it is expressed in the other subgroups 10/13 times (i.e. 77% of YAP1 positive cases are in other subgroups), but expressed without the other markers only 3/13 times (i.e. 23% of YAP1 cases were independent of other lineage markers). This is clear evidence that YAP1 does not define a group independent of the other three markers.

From the Qu et al JTO 2021 paper,

Similarly, in the Baine et al JTO paper (2020), they state "YAP1 was expressed at low levels, primarily in combined SCLC, and was not exclusive of other subtypes." From figure 5 of their paper (below), they show that in YAP1 positive tumors, 63% expressed ASCL1, 42% expressed NeuroD1, and 21% expressed POU2F3. It is unequivocal that YAP1 does not define a distinct group separate from NeuroD1, ASCL1, or POU2F3. (Actually, YAP1 is most enriched in the combined histology group- 100% of combined histology had high YAP1).

YAP1 is likely an important gene in SCLC and may correlate with important molecular features or be a biomarker on its own but this manuscript should not present it as defining a separate and distinct group. This subgroup was initially proposed in a review but subsequent data (including from the authors who initially proposed a YAP1 group) make clear (not merely "controversial") that it is not a distinct group.

RESPONSE TO REVIEWER #4

Reviewer #4 (Remarks to the Author):

In the second revision of the manuscript by Lissa et al, and their second rebuttal, they have reasonably addressed most of the issues I previously raised, but there remain one moderately important and one very important issue that have not been adequately addressed.

Point 1: The moderately important issue is that, as raised earlier, the combination of PARPi/ICB (or topotecan/ATR inhibitor) should be presented separately from PD1 monotherapy or ATR monotherapy; all of these drugs have monotherapy activity (including PARP inhibitors) and predictors of the combination may or may not predict either monotherapy (especially since PARPi and ICB are thought to have different predictors).

Our response: Our rationale for combining patients who received ICI as a single-agent or in combination with a PARPi, is based on the limited efficacy of PARPi in SCLC as monotherapy¹ and the lack of synergistic interaction of PARPi and ICI^{2,3}. The following sentence was added to the limitations in the discussion:

(page 17, lines 479-481): *‘Secondly, the outcomes of the ICB treated cohorts may have been confounded by combination therapy with PARP inhibitor, but multiple studies suggest no added benefit to combining PARP inhibitors with ICI.’*

As reported in the second rebuttal, no patients were treated with ATR inhibitor as single-agent.

Point 2: The more important issue not addressed is that YAP1 is not a distinct subgroup separate from established ASCL1, NeuroD1 and, to a lesser extent, Pou2F3 subgroups. There is compelling evidence from multiple independent groups that three markers define predominantly non-overlapping, biologically distinct subgroups (ASCL1, NeuroD1, and POU2F3), although the fourth distinct group has been the subject of some debate. In a review article, (Rudin et al Nat Rev Cancer) it was suggested YAP1 may define a fourth subgroup but in multiple studies investigating this issue since that time from Rudin et al and others (Baine et al, JTO, 2020; Gay et al, Cancer Cell 2021; Qu et al, JTO, 2021) it is clear and unequivocal that, while YAP1 is expressed in some SCLC cases, it does not define a distinct subgroup but rather overlaps with the others.

To illustrate this in more detail: from the recent paper on which Dr. Thomas is an author (Qu et al, JTO, 2021), the frequency of YAP1 staining is below (Supplemental figure S4). Here, YAP1 is expressed without the other markers in only 2.1% of the cases (3/142), but is expressed along with ASCL1, NEUROD1, or POU2F3 in 10 cases (7%). In other words, of the 13 cases of YAP1 expression, it is expressed in the other subgroups 10/13 times (i.e. 77% of YAP1 positive cases are in other subgroups), but expressed without the other markers only 3/13 times (i.e. 23% of YAP1 cases were independent of other lineage markers). This is clear evidence that YAP1 does not define a group independent of the other three markers.

From the Qu et al JTO 2021 paper,

Similarly, in the Baine et al JTO paper (2020), they state “YAP1 was expressed at low levels, primarily in combined SCLC, and was not exclusive of other subtypes.” From figure 5 of their paper (below), they show that in YAP1 positive tumors, 63% expressed ASCL1, 42% expressed NeuroD1, and 21% expressed POU2F3. It is unequivocal that YAP1 does not define a distinct group separate from NeuroD1, ASCL1, or POU2F3. (Actually, YAP1 is most enriched in the combined histology group- 100% of combined histology had high YAP1).

YAP1 is likely an important gene in SCLC and may correlate with important molecular features or be a biomarker on its own but this manuscript should not present it as defining a separate and distinct group. This subgroup was initially proposed in a review but subsequent data (including from the authors who initially proposed a YAP1 group) make clear (not merely “controversial”) that it is not a distinct group.

Our response: While it is true that Qu et al. reported a low frequency of tumors uniquely positive for YAP1, we would like to point out that the two cohorts are very different. In the study from Qu et al, i) all specimens were primary tumor resections (vs. 15% of biopsies at diagnosis in our study), and ii) every patient was of Chinese ethnicity (vs. one Asian patient in our study). Given that therapy was shown to promote a non-NE phenotype and the genetic variation among ethnic groups, the difference in the prevalence of YAP-1 positive tumors between the two cohorts may be partly explained by these distinct characteristics. As noted in the previous rebuttal, contrary to the studies from Baine et al. and Gay et al., which reported the absence or minimal expression of YAP1 in mainly surgically resected SCLC tumors, YAP1 is uniquely expressed in a subset of tumors in our cohort as shown in Fig 2g and S8b.

We toned down our conclusion about the YAP1 subgroup by adding the following sentence to the discussion:

(page 15, lines 434-437): *‘The role of YAP1 as a subtype-defining marker remains unclear. Although our results are concordant with a recent publication reporting unique YAP1 expression in a subset of SCLCs, other studies suggest that YAP1 does not define a distinct subgroup.’*

References:

1. de Bono, J. *et al.* Phase I, Dose-Escalation, Two-Part Trial of the PARP Inhibitor Talazoparib in Patients with Advanced Germline BRCA1/2 Mutations and Selected Sporadic Cancers. *Cancer Discov* **7**, 620-629 (2017).
2. Thomas, A. *et al.* Durvalumab in Combination with Olaparib in Patients with Relapsed SCLC: Results from a Phase II Study. *J Thorac Oncol* **14**, 1447-1457 (2019).
3. Takahashi, N., Surolia, I. & Thomas, A. Targeting DNA Repair to Drive Immune Responses: It's Time to Reconsider the Strategy for Clinical Translation. *Clin Cancer Res* **26**, 2452-2456 (2020).